# Uncertainty-aware Reward Design Process

**Yang Yang**                                                                                       *yangyang2025@ia.ac.cn*
*National Key Laboratory of Cognition and Decision Intelligence for Complex Systems, Institute of Automation, Chinese Academy of Science*

**Xiaolu Zhou**                                                                          *202321130108@mail.bnu.edu.cn*
*School of Mathematical Sciences, Beijing Normal University*

**Bosong Ding**                                                                           *B.Ding_3@tilburguniversity.edu*
*Air-Lab, Tilburg University*

**Miao Xin**[*]                                                                                         *miao.xin@ia.ac.cn*
*National Key Laboratory of Cognition and Decision Intelligence for Complex Systems, Institute of Automation, Chinese Academy of Science*

**Reviewed on OpenReview:** *https://openreview.net/forum?id=CId5tW1HxR*

## Abstract

Designing effective reward functions is a cornerstone of reinforcement learning (RL), yet it remains a challenging process due to the inefficiencies and inconsistencies inherent in conventional reward engineering methodologies. Recent advances have explored leveraging large language models (LLMs) to automate the design of reward functions. However, LLMs' insufficient numerical optimization capabilities often result in suboptimal reward hyperparameter tuning, while non-selective validation of candidate reward functions leads to substantial computational overhead. To address these challenges, we propose the Uncertainty-aware Reward Design Process (`URDP`), a novel framework that integrates large language models to streamline reward function design and evaluation. `URDP` quantifies candidate reward function uncertainty based on the self-consistency analysis, enabling simulation-free identification of ineffective reward components while discovering novel ones. Furthermore, we introduce uncertainty-aware Bayesian optimization (UABO), which incorporates uncertainty estimation to improve the hyperparameter configuration. Finally, we construct a bi-level optimization framework by decoupling the reward component optimization and the hyperparameter tuning. `URDP` promotes the collaboration between the reward logic reasoning of the LLMs and the numerical optimization strengths of the Bayesian optimization. We conduct a comprehensive evaluation of `URDP` across 35 diverse tasks spanning three benchmark environments: Isaac-Gym, Bidexterous Manipulation, and ManiSkill2. Our experimental results demonstrate that `URDP` not only generates higher-quality reward functions but also achieves significant improvements in the efficiency of automated reward design compared to existing approaches. We open-source all code at `https://github.com/Yy12136/URDP`.

## 1 Introduction

In reinforcement learning (RL), the design of reward functions serves as a pivotal determinant for successfully training agents in sequential decision-making tasks. These rewards guide the learning process by shaping agent behaviors to accomplish complex objectives across diverse environments. While conventional approaches such as reward engineering and inverse reinforcement learning (IRL) (Arora & Doshi, 2021) established early research paradigms, they remain fundamentally constrained by their reliance on human expertise and the

---

[*]Corresponding authors.

availability of high-quality demonstration data, particularly in domains like robotic skill acquisition (Zitkovich et al., 2023). Recent advancements in large language models (LLMs) (Radford et al., 2019; Brown et al., 2020; Liu et al., 2024) have demonstrated remarkable capabilities in natural language understanding, code generation, and contextual optimization, thereby introducing a novel paradigm for automated reward function design.

However, current automated reward design methodologies based on LLMs present two fundamental challenges (Cao et al., 2024). First, the **efficiency** of reward function design remains suboptimal. Existing approaches rely heavily on simulation-based training processes (Ma et al., 2024a) to evaluate reward function efficacy, which involves extensive and often redundant evaluations, leading to significant computational overhead. Second, the **performance** of LLM-generated reward functions frequently falls short of expectations. During the optimization process, LLMs fail to fully leverage their reasoning capabilities, resulting in reward functions that inadequately capture the intended task objectives. Given that the design efficiency and the performance of reward functions are closely tied to the speed and effectiveness of policy learning, a key research question emerges: *How can we enhance the efficiency of obtaining performant reward functions?*

In this paper, we introduce the **Uncertainty-aware Reward Design Process** (`URDP`), a novel framework for automated reward function generation. First, we propose a method to quantify the uncertainty of generated samples, which enables the selective elimination of redundant reward function sampling and simulation. This approach is grounded in a key observation regarding self-consistency (Wang et al., 2022): LLMs exhibit higher output consistency when handling well-defined tasks, allowing for more efficient sampling strategies. Second, we identify a critical limitation in current LLM-only evolutionary search approaches, i.e., their suboptimal performance in numerical optimization. To address this, we decouple reward component formulation from reward intensity optimization, delegating the latter to a dedicated numerical optimization module. Specifically, we propose a novel uncertainty-aware Bayesian optimization (Snoek et al., 2012) approach incorporating uncertainty distribution priors, which significantly accelerates convergence in this black-box optimization task. Experimental results demonstrate that `URDP` surpasses state-of-the-art methods in both reward performance and computational efficiency, establishing a new benchmark for automated reward design.

Overall, our contributions are summarized as follows: (1) We propose `URDP`, a novel framework for automated reward function design in reinforcement learning. The framework employs an alternating bi-level optimization process that decouples reward component design from hyperparameter optimization, thereby significantly enhancing reward function performance. (2) We introduce a self-consistency-based reward uncertainty quantification method for the reward component design process. This approach significantly improves the efficiency of the reward component validation. (3) We present an Uncertainty-aware Bayesian optimization algorithm that substantially increases the efficiency of hyperparameter search. (4) Our comprehensive evaluation across 35 tasks spanning 3 distinct benchmarks demonstrates that `URDP` consistently outperforms existing methods in both reward function generation efficiency and final policy performance, as evidenced by rigorous quantitative analysis.

## 2 Related work

**Reward code generation.** The automation of reward code generation has been a critical area of research, aiming to simplify and improve the process of defining task-specific reward functions for RL. As the dual formulation of RL problems, inverse reinforcement learning (IRL) methods (Arora & Doshi, 2021) have been extensively investigated for reward function acquisition. However, these approaches are fundamentally limited by their dependence on demonstration data, which severely constrains their scalability in practical applications. Recently, LLM-based reward design methodologies (Kwon & Michael, 2023; Yu et al., 2023; Ma et al., 2024a; Xie et al., 2024) have demonstrated promising potential, offering a paradigm shift in automated reward function development. L2R (Yu et al., 2023) introduced a two-stage LLM-prompting framework to generate templated rewards, bridging high-level language instructions with low-level robot actions. Eureka (Ma et al., 2024a) leveraged the zero-shot and in-context learning capabilities of advanced LLMs to perform evolutionary optimization over the reward code. This method demonstrated the potential of LLMs to generate rewards without task-specific prompting or predefined templates, enabling agents to acquire complex skills via RL. Text2Reward (Xie et al., 2024) extended this line of work by generating shaped,

dense reward functions as executable programs grounded in compact environment representations. Unlike sparse reward codes or constant reward functions, Text2Reward produces interpretable, dense reward codes capable of iterative refinement with human feedback. Despite their success, these methods generally do not separate the tasks of defining reward components and tuning parameters, which may reduce the efficiency of automated reward design. Our proposed framework decouples diverse factors in reward design to reduce invalid tuning. It incorporates uncertainty quantification to guide LLMs toward more focused analysis and refinement of reward logic relevance.

**Hybrid optimization.** Recent advances in large language models (LLMs) (OpenAI, 2023; Liu et al., 2024) have demonstrated remarkable progress in text-based complex reasoning tasks (Li et al., 2025). Through techniques such as self-improvement (Song et al., 2023), multi-path reasoning (Wan et al., 2024) and reward modeling (Zhong et al., 2025), LLMs exhibit substantial potential in contextual comprehension (OpenAI, 2023), code generation (Yu et al., 2024) and task planning (Hao et al., 2023). However, their capabilities in deep logical reasoning (Cheng et al., 2025), particularly in mathematical and numerical optimization domains (Yan et al., 2025), remain underexplored, with significant performance gaps persisting. Several studies attempt to employ LLMs as meta-optimizers for diverse optimization problems (Yang et al.). Yet, due to their inherently discrete representation nature, LLMs' effectiveness in high-dimensional, continuous numerical reasoning tasks requires further investigation (Assran et al., 2025). In reinforcement learning, the reward design inherently involves multiple types of optimization problems. Unlike completely LLM-only evolutionary search approaches (Ma et al., 2024b; Xie et al., 2024) that utilize LLM-based evolutionary search to optimize all elements in the reward function, *URDP* only applies it to the optimization of reward components, while a black-box numerical optimization tool is introduced to compensate for the limitations of LLMs in continuous numerical optimization.

**Uncertainty quantification.** Uncertainty quantification (UQ) serves as a fundamental component for reliable automated decision-making and has been extensively studied in domains such as Bayesian inference (Gal & Ghahramani, 2015; Foong et al., 2020). Recent advances have investigated UQ in black-box language models (Liu et al., 2025; Geng et al., 2024; Kuhn et al., 2023; Lin et al.), yielding various approaches including token-level entropy methods (Kadavath et al., 2022), conformal prediction-based techniques (Su et al., 2024), and consistency-based frameworks (Lin et al.). While existing methods primarily leverage uncertainty estimation to enhance LLM interpretability (Ahdritz et al., 2024) and mitigate hallucination risks (Shorinwa et al., 2024; Mohri & Hashimoto, 2024), our methodology not only actively quantifies the uncertainty in LLMs but also strategically leverages this uncertainty as the foundational mechanism for both reward function design and efficient numerical optimization. Furthermore, we identify and characterize a significant correlation between the novel reward component and its uncertainty.

## 3 Preliminary: problem setup and notations

**Reinforcement learning (RL)** tasks can be modeled as Markov Decision Processes (MDPs) defined by the tuple $\langle \mathcal{S}, \mathcal{A}, \mathcal{P}, \mathcal{R}, \gamma \rangle$, where $\mathcal{S}$ is the state space, $\mathcal{A}$ is the action space, $\mathcal{R}$ is the space of reward functions, $R \in \mathcal{R}$ is the reward function, and $\gamma$ is the discount factor. $P(\mathbf{s}_{t+1}|\mathbf{s}_t, \mathbf{a}_t) \in \mathcal{P}$ is the transition probability of moving to state $\mathbf{s}_{t+1}$ given the current state $\mathbf{s}_t$ and action $\mathbf{a}_t$, where $\mathcal{P}$ is the transition probability space. The agent can generate a trajectory of state and action $\tau = (\mathbf{s}_0, \mathbf{a}_0, \mathbf{s}_1, \mathbf{a}_1, ...)$ by executing a policy $\pi(\mathbf{a}_t|\mathbf{s}_t)$ in the policy space $\Pi$. The goal is to find the optimal policy that maximizes the sum of expected returns:

$$J(\pi) = \mathbb{E}_{\tau \sim \pi} \left[ \sum_{t=0}^{T} \gamma^t R(\mathbf{s}_t, \mathbf{a}_t) \right]. \tag{1}$$

Following the previous works (Song et al., 2023; Ma et al., 2024a; Xie et al., 2024), the reward function $R$ is represented as a function of the *reward components* ($\mathbf{r}$) and the *reward intensities* ($\theta$)

$$R(\mathbf{s}, \mathbf{a}) = h(\mathbf{r}, \theta, \mathbf{s}, \mathbf{a}), \tag{2}$$

where the function $h$ and the reward components $\mathbf{r}$ are represented in code form. Here, the reward components $\mathbf{r}$ are defined as the composable and interpretable sub-terms that constitute the overall reward

function Escontrela et al. (2022), each measuring a specific behavioral aspect or task objective. For instance, `distance_reward` measures the agent's proximity to the target, while `velocity_reward` quantifies the alignment of its velocity toward the goal direction. These components are implemented as modular units, enabling both structural manipulation and numerical tuning during the optimization process (see the highlighted red text in the example of Appendix D.1). The reward intensity refers to the weight of a reward component within the total reward. Designing reward functions represented in code form is critical to aligning agent behavior with task objectives. Here, the reward design problem is defined as follows.

**Definition 1: Reward design problem (RDP)** (Singh et al., 2009). A reward design problem is represented as $RDP = \langle M, \mathcal{R}, \pi_M, F \rangle$, where $M = (S, A, P)$ is the environment model. $\mathcal{R}$ is the space of reward functions and $\Pi$ is the space of polices. $\mathcal{A}_M(\cdot) : \mathcal{R} \rightarrow \Pi$ is an algorithm to optimize the reward function $R \in \mathcal{R}$. $F : \Pi \rightarrow \mathbb{R}$ is the fitness function that produces a scalar evaluation of the RL policy $\pi : S \rightarrow A$ using the ground truth reward function. The goal of an RDP is to generate a reward function $R \in \mathcal{R}$ using $\mathcal{A}_M(R)$ to achieve the highest fitness score $F(\mathcal{A}_M(R))$.

Evaluating the performance of a designed reward function necessitates computationally expensive simulations (training a policy $\pi$ using the designed $R$). Notably, the original definition of RDP does not impose constraints on sample efficiency. In this work, our objective is to output a reward function code $R$ for a given task context such that $F(\mathcal{A}_M(R))$ is maximized while maximizing sample efficiency throughout the automatic reward design process.

**Bayesian optimization (BO)** is a sequential design strategy for global optimization of black-box functions (Shahriari et al., 2015; Snoek et al., 2012). Given a black-box function $f : \mathbb{X} \rightarrow \mathbb{R}$, Bayesian optimization aims to find an input $\mathbf{x}^* \in argmin_{x \in \mathbb{X}} f(\mathbf{x})$ that globally minimizes $f$. It places a prior $p(f)$ over the objective function $f$ to form a surrogate model (usually a Gaussian process (Wang et al., 2024)). An *acquisition function* (e.g., the Expected Improvement (EI) (Ament et al., 2023)) $a_{p(f)} : \mathbb{X} \rightarrow \mathbb{R}$ strategically determines the direction of the search for sampling points. The algorithm iteration proceeds in the following three steps: (1) find the most promising $\mathbf{x}_{n+1} \sim argmax \ a_p(\mathbf{x})$; (2) evaluate the function $y_{n+1} = f(\mathbf{x}_{n+1})$ and update the set of historical observations $\mathcal{D}_n = (x_j, y_j)_{j=1...n}$ by adding the point $(\mathbf{x}_{n+1}, y_{n+1})$, and (3) update $p(f|\mathcal{D}_{n+1})$ and $a_{p(f|\mathcal{D}_{n+1})}$.

## 4 Methods

The Uncertainty-aware Reward Design Process (`URDP`) framework incorporates three fundamental elements: (1) Decoupling of the reward component and reward intensity design processes, (2) Reward component generation based on uncertainty quantification, and (3) Uncertainty-aware Bayesian optimization. `URDP` enhances the efficiency of reward design by minimizing redundant simulations while improving the performance of generated reward functions through the integration of numerical optimization techniques within the decoupled optimization framework.

### 4.1 Decoupled Reward Generation and Hyperparameter Optimization

Large language models possess extensive commonsense knowledge about task rewards, enabling them to surpass human-level performance in designing reward components (Yu et al., 2023; Ma et al., 2024a; Xie et al., 2024) for certain tasks. However, their capability in black-box numerical optimization remains inferior to specialized numerical optimization algorithms (see Section 5.4 Abl-3 for an ablation study). Consequently, existing methods (Ma et al., 2024a) that conflate the optimization of reward components with reward intensities not only yield suboptimal numerical optimization results but also lead to insufficient optimization of the reward components.

**URDP framework**. Building upon these observations, we propose a decoupled reward function design process. As illustrated in Figure 1, `URDP` implements a bi-level iterative optimization procedure. Given the environment specifications and task descriptions, the agent first samples multiple reward functions from the LLM in the outer loop. It ranks them using uncertainty quantification metrics and filters out redundant or potentially unreliable rewards through a process termed *Reward Code Sampling with Uncertainty Screening*. Subsequently, the agent invokes the proposed numerical optimization tool (UABO) in the inner loop to determine optimal

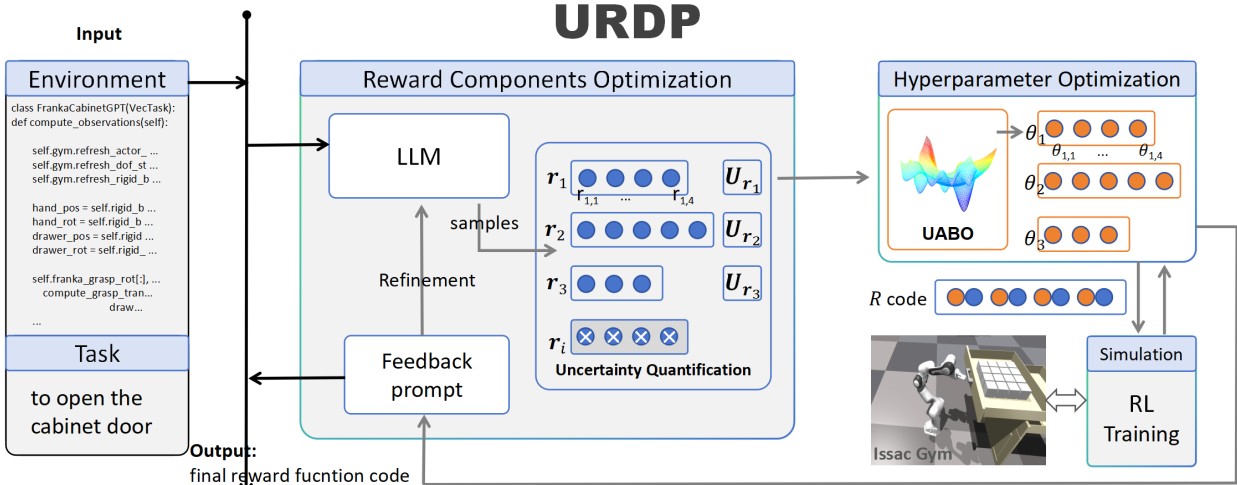

Figure 1: *URDP* implements an alternating bi-level iterative optimization framework for automated reward design problems (RDP). The outer-loop optimization employs LLMs to refine reward components, where uncertainty quantification significantly enhances sampling efficiency. Concurrently, the inner-loop optimization utilizes Uncertainty-Aware Bayesian Optimization (UABO) to determine optimal hyperparameter configurations for the reward components. The decoupled architecture strategically leverages the complementary strengths of LLMs in conceptual reward design and numerical optimization tools in precise parameter tuning, achieving synergistic improvements in both final policy performance and computational efficiency.

reward intensity hyperparameters for the current reward configuration through black-box optimization and simulation-based evaluation. Finally, the agent evaluates the feasibility of the current reward components and constructs a feedback prompt to refine them. The feedback available to the process is encapsulated in a structured prompt sent to the LLM. This prompt involves task-level indicators such as success rate and episode length, component-level statistics (mean, max, and min values of each reward component), sample-level statistics (uncertainty scores, standard deviation, and range), information about the best-performing reward functions, and adjustment suggestions for refining the reward design. An example of the feedback information, along with the full templates of both the initial and feedback prompts, is provided in App. A. In essence, the outer loop optimizes the reward components while the inner loop tunes reward intensity hyperparameters, with their alternating optimization progressively converging toward optimal reward functions. See Alg. 1 for pseudocode.

## 4.2 Reward Code Sampling with Uncertainty Screening

In LLM-based RDP, automated reward function design can be achieved through iterative sampling and simulation. However, existing approaches indiscriminately conduct simulation-based evaluation on all LLM-generated reward function samples, despite the frequent presence of redundant or infeasible candidates within these samples. Our analysis identifies this as a critical factor contributing to computational inefficiency (see results in Section 5.4 Abl-1). Consequently, an essential question emerges: *how to effectively filter out potentially redundant reward function samples before simulations*, thereby avoiding computationally expensive yet unnecessary simulation training.

**Uncertainty priority**. Our sampling approach is grounded in the principle of *self-consistency* (Wang et al., 2022) in LLMs. When explicitly prompted to generate diverse outputs, LLMs that produce highly consistent responses demonstrate well-internalized, task-specific knowledge. Such outputs exhibit high reliability and typically require minimal refinement. Conversely, if the generated results show substantial diversity, this indicates uncertainty in the understanding of the task context and underlying concepts. These divergent results exhibit lower reliability and consequently demand more refinement.

**Sampling and filtering**. Based on this principle, the agent prompts the LLM to generate diverse reward components for a given RL task. The uncertainty score of each reward component for the reward function is quantified by its occurrence frequency across all sampled candidates. To quantify the uncertainty of the reward component, `URDP` identifies and resolves ambiguities in reward components using LLMs. It combines textual similarity and semantic similarity analyses to evaluate the relevance and clarity of reward components, assigning the reward component uncertainty score $U_{r_{i,m}}$ to each reward component $r_{i,m}$ as

$$U_{r_{i \in K, m \in M_i}} = 1 - \sum_{i \in [1,K]} \left( u \left( max(S_{\text{text}}(r_{i,m}), S_{\text{semantic}}(r_{i,m})) - \omega \right) \right) / K, \qquad (3)$$

where $u(\cdot)$ is a step function, $K$ denotes the quantity of the reward function samples, $M_i$ is the number of reward components in the reward function $R_i$, $\omega$ is a decision parameter regarding the maximum similarity ($\omega = 0.95$), $S_{\text{text}} \in (0,1]$ and $S_{\text{semantic}} \in (0,1]$ are the textual and semantic similarity scores, respectively. Furthermore, the normalized sample uncertainty score $U_{R_i}$ is computed to evaluate the overall uncertainty of each reward function sample $R_i$. Implementation details and hyperparameter settings are elaborated in App. C. Using $U_{R_i}$, the agent filters out samples containing identical reward components, thereby eliminating redundant inner-loop optimization processes that would otherwise incur unnecessary computational overhead. Moreover, our analysis reveals that highly uncertain reward components may contain unexplored components capable of facilitating effective reward shaping (see Section 5.5 Disc-2). Consequently, the agent implements an adaptive exploration-exploitation strategy, i.e., using $N_{inner} \cdot U_{R_i}$ to adjust the maximum number of iterations in the inner loop, where $N_{inner}$ is the upper limit. For high-uncertainty samples, it allocates additional inner-loop iterations to prioritize exploration of optimal hyperparameter configurations for potentially novel reward components. For low-uncertainty samples, it emphasizes exploitation to minimize unnecessary simulations. This approach balances the trade-off between the exploration and utilization of uncertain reward components.

### 4.3 Uncertainty-aware Bayesian Optimization

While large language models demonstrate significant potential for reward component design, they exhibit suboptimal performance in numerical optimization tasks (see results in Section 5.4 Abl-3). This limitation leads to non-optimal reward intensity configurations in LLM-generated reward functions, representing a key factor in the poor policy learning performance observed in prior approaches. In contrast to existing methods, our `URDP` framework does not rely on LLM-based agents for direct numerical optimization. Instead, the agent serves as a controller that coordinates specialized numerical optimization tools. Specifically, `URDP` delegates the inner-loop optimization of reward intensity parameters to Bayesian optimization (BO) algorithms. Benefiting from BO's superiority in black-box global optimization, this approach achieves significantly better performance than LLM-based optimization.

Although the classical Bayesian optimization algorithm, *i.e.*, Gaussian Process with Expected Improvement (EI) (Ament et al., 2023; Snoek et al., 2012), demonstrates theoretical advantages, its practical efficiency remains unsatisfactory, particularly due to the substantial computational overhead incurred by acquisition functions during sampling (simulation training). This inefficiency frequently prevents convergence to globally optimal solutions within an acceptable number of samplings. Notably, the uncertainties of individual reward components imply valuable prior knowledge for enhancing BO's sampling efficiency. Given a reward function comprising $m$ components, we model the $m$ reward intensities as a joint probability distribution. Crucially, higher uncertainty in $\{r_{i,1}, ..., r_{i,m}\}$ corresponds to a more uniform marginal distribution along that dimension. This observation suggests that sampling should prioritize exploitation over exploration in high-uncertainty dimensions. Building upon this smoothness assumption, we propose **Uncertainty-aware Bayesian optimization (UABO)** to address these limitations.

**UABO** incorporates reward component uncertainty scores, $\mathbf{U}_{\mathbf{r}_i} = \{U_{r_{i,1}}, ..., U_{r_{i,m}}\}$, into both the kernel function and the acquisition function of the standard Bayesian optimization. The Matern kernel in the Gaussian process has the form

$$k(p, p') = f_\nu(d) = \sigma^2 \cdot \frac{2^{1-\nu}}{\Gamma(\nu)} \left( \frac{\sqrt{2\nu}d}{\ell} \right)^\nu K_\nu \left( \frac{\sqrt{2\nu}d}{\ell} \right), \qquad (4)$$

where $d$ is the Euclidean distance between $p$ and $p'$, $\sigma^2$ is the variance, $\nu$ is the smoothness parameter, $\ell$ is the length scale parameter and $K_\nu$ is the modified Bessel function of the second kind. We note that the kernel is isotropic, which means that all dimensions (i.e., the intensity parameters of reward components) share the same length scale parameter. To accommodate heterogeneous smoothness (uncertainty score $U(r_i)$) across different dimensions, we propose an anisotropic kernel function that incorporates uncertainty values as length scales within the distance metric. The distance is formulated as follows,

$$d_{\mathrm{u}}(\theta_i^{t-1}, \theta_i^t) = \sqrt{\left(\frac{\theta_{i,1}^{t-1} - \theta_{i,1}^t}{U_{r_{i,1}}}\right)^2 + \cdots + \left(\frac{\theta_{i,m}^{t-1} - \theta_{i,m}^t}{U_{r_{i,m}}}\right)^2}, \tag{5}$$

where $\theta_i^{t-1}$ and $\theta_i^t$ are reward intensity hyperparameters for the reward components $\mathbf{r}_i$ in two iterations. Then the new kernel function is defined as

$$\tilde{k}(\theta_i^{t-1}, \theta_i^t) = f_\nu(d_{\mathrm{u}}) = \sigma^2 \cdot \frac{2^{1-\nu}}{\Gamma(\nu)} \left(\frac{\sqrt{2\nu} d_{\mathrm{u}}}{U_{R_i}}\right)^\nu K_\nu \left(\frac{\sqrt{2\nu} d_{\mathrm{u}}}{U_{R_i}}\right). \tag{6}$$

Furthermore, we leverage $\mathbf{U}_{\mathbf{r}_i}$ to enhance the performance of the acquisition function. The standard Expected Improvement (EI) acquisition function (Ament et al., 2023) in Bayesian Optimization is as follows,

$$\mathrm{EI}_{y^\star}(x) = \mathbb{E}_{f(x) \sim \mathcal{N}(\mu(x), \sigma^2(x))} \left[[f(x) - y^\star]_+\right] = \sigma(x) h\left(\frac{\mu(x) - y^\star}{\sigma(x)}\right), \tag{7}$$

where $[\cdot]_+ = \max(0, \cdot)$, $y^\star = \max_i y_i$ is the best observed value, and $h(z) = \phi(z) + z\Phi(z)$, $\phi$ is the standard normal distribution density and $\Phi$ is the distribution function. In our context, $y$ is the agent success rate from the simulation RL training, and $x$ is the reward intensities $\theta_i$. The objective of EI is to decide the next promising reward intensity parameters for the next round of simulation.

To reduce inefficient exploration along directions with potentially insignificant influence on the function value, we introduce a penalty term that constrains the weighted distance between the candidate point and the current optimum, yielding an uncertainty-accelerated EI acquisition function (uEI) defined as

$$\mathrm{uEI}_{y^\star}(\theta_i, \mathbf{U}_{\mathbf{r}_i}) = \mathrm{EI}(\theta_i) \cdot w(\theta_i), \tag{8}$$

$$w(\theta_i) = \exp\left(-\sum_{m=1}^{M_i} U_{r_{i,m}}(\theta_i - \theta_i^*)^2\right), \tag{9}$$

where $w(\theta_i)$ is a penalty term to constrain the weighted distance between $\theta_{\mathbf{i}}$ and the current best $\theta_i^\star$. Specifically, an uncertainty value approaching zero for a particular dimension indicates no restriction on variations along that dimension. Conversely, a large uncertainty weight in a certain direction implies that extensive exploration in that direction is discouraged. UABO demonstrates significantly improved convergence efficiency (see Section 5.4 Alb-3), reaching optimal values within limited hyperparameter search steps, thereby substantially enhancing the performance and effectiveness of reward intensity configuration. A formal proof of its convergence lower bound is provided in App. F.

## 5 Experiments

In this section, we evaluate the proposed `URDP` framework through extensive experiments on a diverse set of environments and tasks, comparing its performance against human and baseline approaches. All experiments and comparative analyses presented in this paper utilize DeepSeek-v3-241226 (Liu et al., 2024) as the foundational model unless explicitly stated otherwise (see App. G.2 for more open-source LLMs' results).

### 5.1 Baselines

**Eureka** (Ma et al., 2024a) (Baseline) provides a systematic approach for generating reward functions utilizing LLMs. It incorporates feedback from various evaluation results to refine the reward functions via evolutionary iterations. This iterative process continues until an optimal reward function is achieved.

---

**Algorithm 1:** Uncertainty-aware Reward Design Process

---

**Input:** Task description $T$, Environment code $E$, LLM $\mathcal{L}$.
**Output:** Optimized reward function $R^*$.
**foreach** *iteration* $n \in N_{outer}$ **do**

    Generate $K$ reward component samples $\mathbf{r}_i = \{r_{i,1}, r_{i,2}, \ldots, r_{i,m}\}_{i \in K}$ using $\mathcal{L}(T, E, prompt)$

    Uncertainty quantification: $\mathbf{U}_{\mathbf{r}_i} = \{U_{r_{i,1}}, \ldots, U_{r_{i,m}}\}$ and $U_{R_i}$

    Filter out redundancies and reserve $R_{i \in K^*}$

    **foreach** $R_i$, $t = 1, 2, \ldots N_{inner} \cdot U_{R_i}$ **do**

        Fit probabilistic model for $f(\theta_i)$ on data $D_{i,t-1}$

        Choose $\theta_i^t$ by maximizing the acquisition function $uEI(\theta_i, \mathbf{U}_{\mathbf{r}_i})$

        Evaluate by simulation training $y_{i,t} = f(\theta_i^t)$

        Augment the data $D_{i,t} = D_{i,t-1} \cup (\theta_i^t, y_{i,t})$

        Choose incumbent $\theta_i^* \leftarrow argmax\{y_{i,1}, \ldots, y_{i,t}\}$ and $y_i^* \leftarrow max\{y_{i,1}, \ldots, y_{i,t}\}$

    Construct a feedback prompt for LLM to propose revised components

Choose optimal $\{(\mathbf{r}^*, \theta^*)\} \leftarrow argmax\{y_1^*, \ldots, y_{K^*}^*\}$
Recombine $\{(\mathbf{r}^*, \theta^*)\}$ into $R^*$
**return** $R^*$.

---

**Text2reward** (Xie et al., 2024) is a reinforcement learning method that automatically generates dense reward functions from natural language task descriptions using LLMs, without relying on expert data or demonstrations, and can express human goals in the form of procedural rewards, given to iterations using human feedback.

**Human**. To maintain a fair comparison, we adopted the same Human data as reported in Eureka (Ma et al., 2024a). The original shaped reward functions provided in the benchmark tasks were developed by active reinforcement learning researchers who designed the tasks. These reward functions embody the outcomes of expert human reward engineering.

**Sparse**.These functions correspond to the fitness measures $F$ employed to assess the performance of the generated reward signals. Analogous to human feedback, they are also provided as part of the benchmark suite. The detailed configurations for Dexterity tasks and Isaac tasks are consistent with those used in Eureka (Ma et al., 2024a). The fitness functions of all tasks in ManiSkill2 are specified in App. B.2.

## 5.2 Experimental Setup

**Benchmarks**. Our environments consist of three benchmarks: Isaac, Dexterity, and Maniskill2. They comprise 35 different tasks. Nine of these tasks are from the original Isaac Gym environment (Nasir et al., 2024) (Issac), twenty are complex bi-manual tasks (Chen et al., 2022) (Dexterity), and the remaining six are from the Maniskill2 environment (Gu et al.). See App. B for more details.

**Metrics**. We examine fore metrics: i. Success Rate (**SR**). We report the success rates of different reward functions on Dexterity and ManiSkill2 tasks. To ensure fair comparison with the baseline methods, the success rates for ManiSkill2 tasks are calculated using the last 50% of test results from each evaluation, while full test results are used for Dexterity tasks. (ii) Human Normalized Score **(HNS).** For *Isaac* tasks, following EUREKA (Ma et al., 2024a), we use HNS $= \frac{F_{\text{Method}} - F_{\text{Sparse}}}{|F_{\text{Human}} - F_{\text{Sparse}}|}$, where $F$ is the environment's fitness functions (e.g., binary success on Bidexterous tasks; continuous task score on IsaacGym tasks). Scores are normalized so that Sparse maps to 0 and Human to 1, enabling cross-task comparison. An HNS$> 1$ indicates the generated reward's training result exceeding the human baseline. iii. The Number of Evaluations (**NOE**), quantified as the total number of simulations (RL training using the designed candidate reward functions) conducted across all samples during the optimization process. iiii. The Number of LLM Callings (**NLC**), representing the cumulative number of LLM calls made during both reward function generation and refinement. These metrics collectively provide a comprehensive assessment, with SR and HNS evaluating the performance of the generated reward functions, and NOE and NLC quantifying the efficiency of the design process.

**Policy Learning**. The performance of reward functions generated by *URDP* and comparative methods was rigorously validated through RL training. For both Isaac and Dexterity environments, we employ the same high-efficiency PPO (Schulman et al., 2017) implementation as used in Eureka, using identical task-specific hyperparameters without modification. In the ManiSkill2 environment, we utilized both SAC (Haarnoja et al., 2018) and PPO algorithms to ensure fair comparison between *URDP*, Text2Reward, and Eureka, strictly maintaining the original hyperparameter configurations across all methods. See App. C.2 for detailed parameter configurations.

### 5.3 Results

**URDP improves the efficiency of the reward function design**. Table 1 presents a comprehensive comparison of computational efficiency between *URDP* and Eureka across three benchmarks. When optimizing for the peak reward performance, *URDP* requires only 52.4% of the simulation episodes (NOE) and 46.6% of the evolutionary iterations (NLC) compared to Eureka. This significant acceleration demonstrates the superior optimization efficiency of the *URDP* in automated reward function design. See App.E for a per-task breakdown.

Table 1: *URDP* demonstrates superior efficiency across all benchmarks.

| Methods | Isaac NOE↓ | Isaac NLC↓ | Dexterity NOE↓ | Dexterity NLC↓ | ManiSkill2 NOE↓ | ManiSkill2 NLC↓ |
|---|---|---|---|---|---|---|
| Txet2Reward | 72.889 | 5 | 84.45 | 6.05 | 106.667 | 6.667 |
| Eureka | 68.667 | 4.556 | 80.05 | 5.5 | 98.667 | 6.167 |
| *URDP* | **39.501** | **2.495** | **57.8** | **3.4** | **32.33** | **1.667** |

**URDP demonstrates superior reward function performance**. Table 2 presents a systematic comparison of reward functions generated by different approaches across benchmark tasks. Under identical simulation budgets (NOE), reinforcement learning agents trained with *URDP*-derived reward functions achieve significantly higher success rates than competing methods. Notably, *URDP* demonstrates substantial performance gains of 132%, 45%, and 76% over Eureka across the three experimental environments, representing significant enhancements. Furthermore, *URDP*-designed reward functions outperform manually engineered counterparts by a considerable margin, providing compelling evidence for the efficacy of automated RL frameworks.

Table 2: *URDP* exhibits higher reward performance across all benchmarks.

| Methods | Isaac (HNS↑) | Dexterity (SR↑) | ManiSkill2 (SR↑) |
|---|---|---|---|
| Sparse | 0 | 0.054 | 0.101 |
| Human | 1.000 | 0.459 | 0.434 |
| Text2Reward | 1.553 | 0.452 | 0.554 |
| Eureka | 1.607 | 0.466 | 0.449 |
| *URDP* | **3.424** | **0.675** | **0.792** |

**URDP achieves synergistic progress in both reward performance and generation efficiency**. In Figure 2, it can be seen that the superiority of *URDP* over Eureka, text2reward and Human, both in terms of the success rate and the reduction in the number of simulations, has been well improved on these typical tasks. Each data point in the line plots represents the effectiveness of the reward function obtained after a single iteration. The results show that *URDP* surpasses human-designed rewards on certain tasks after just one LLM refinement cycle and achieves optimal performance with significantly fewer iterations than both Eureka and Text2Reward, which means that it consumes fewer tokens. These results collectively demonstrate that *URDP* achieves simultaneous improvements in both the efficiency and performance of the reward generation.

### 5.4 Ablation Experiments

Furthermore, we explore the role of each core content in *URDP* in achieving the above results.

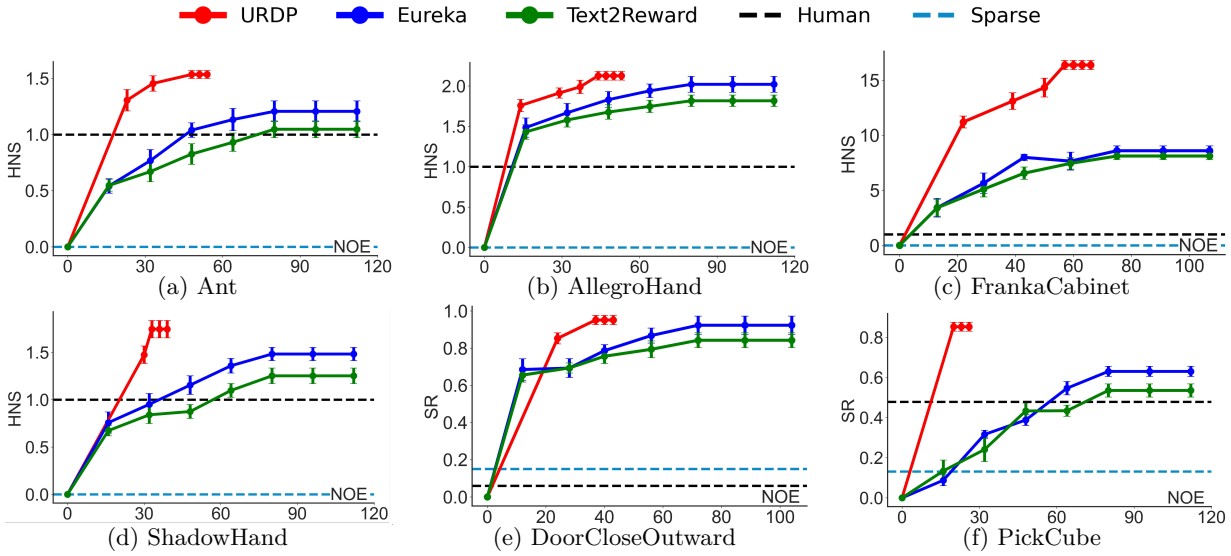

Figure 2: Comparisons of *URDP* with other methods in Isaac (a-d), Dexterity (e), and ManiSkill2 (f). All curves are plotted until their convergence plateau. The HNS values no longer change across simulations (the plot points in the curves) after at least two consecutive outer-loop iterations. At this point, the inner loop (UABO) no longer yields better hyperparameters, as it produces identical results in three consecutive evaluations, and the outer-loop LLM optimization no longer yields improved reward components. The same criterion is applied consistently across all methods and figures.

**Abl-1: Uncertainty quantification improves the efficiency of reward design**. To evaluate the role of uncertainty sampling, we conduct ablation studies by removing the uncertainty sampling and filtering module from *URDP* (denoted as **URDP w.o. Uncertainty**). Experimental results in Figure 3 demonstrate that *URDP* achieves comparable success rates while requiring significantly fewer optimization episodes (NOE) than URDP w.o. Uncertainty, quantitatively validating the efficiency improvement brought by uncertainty-aware sampling. Interestingly, our analysis also reveals the correlation between the uncertainty and the novel reward functions, with detailed mechanistic explanations to be discussed in Section 5.5 (Disc-2).

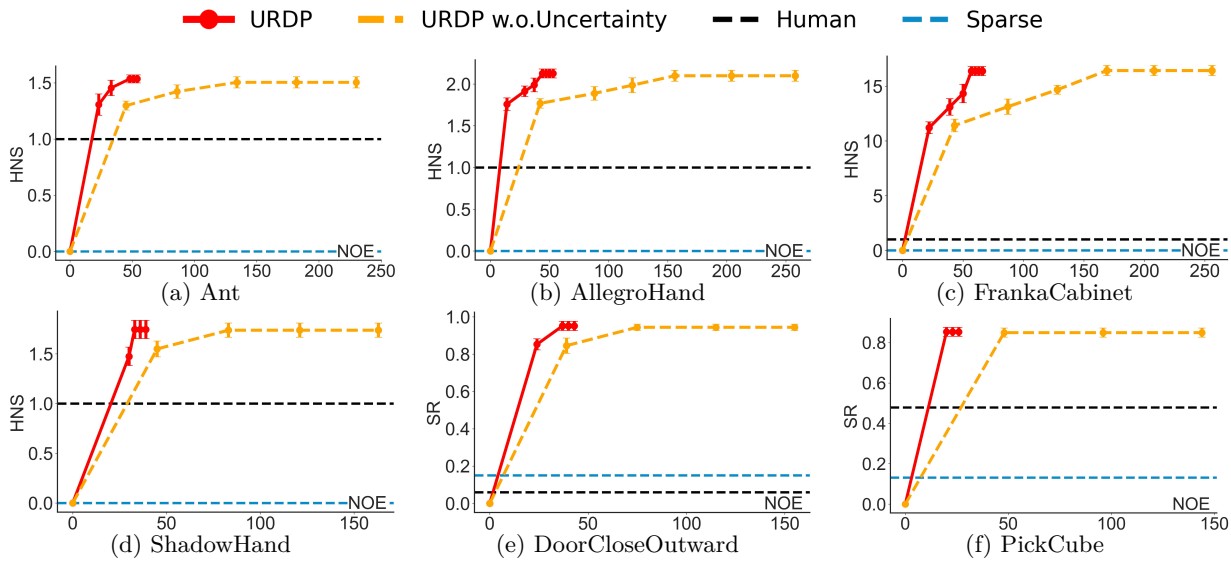

Figure 3: When generating reward functions of comparable performance, *URDP* requires significantly fewer simulation training episodes, attributable to its effective uncertainty-based filtering mechanism.

**Abl-2: Decoupled optimization is the cornerstone for the collaborative improvement of performance and efficiency**. This experimental investigation examines the role of decoupling in *URDP*, where reward components and their associated intensities are optimized separately. To evaluate this mechanism, we ablate UABO from *URDP* (denoted as **URDP w.o. UABO**) while maintaining identical configurations otherwise, resulting in a system where both reward components and intensities are jointly configured by the LLM without alternating optimization. Under unrestricted NLC, we compare the HNS or SR achieved by URDP w.o. UABO using equivalent NOE to the standard *URDP* implementation. As shown in Figure 4 (dashed lines), consistent reductions in both HNS and SR are observed across all three benchmark tasks, demonstrating the substantial impact of decoupled optimization on improving the reward performance. Furthermore, *URDP* exhibits faster convergence (requiring fewer NLC) to optimal solutions, suggesting that decoupling also enhances the efficiency of evolutionary search. A comprehensive discussion of this phenomenon and the underlying mechanisms is presented in Section 5.5 (Disc-1).

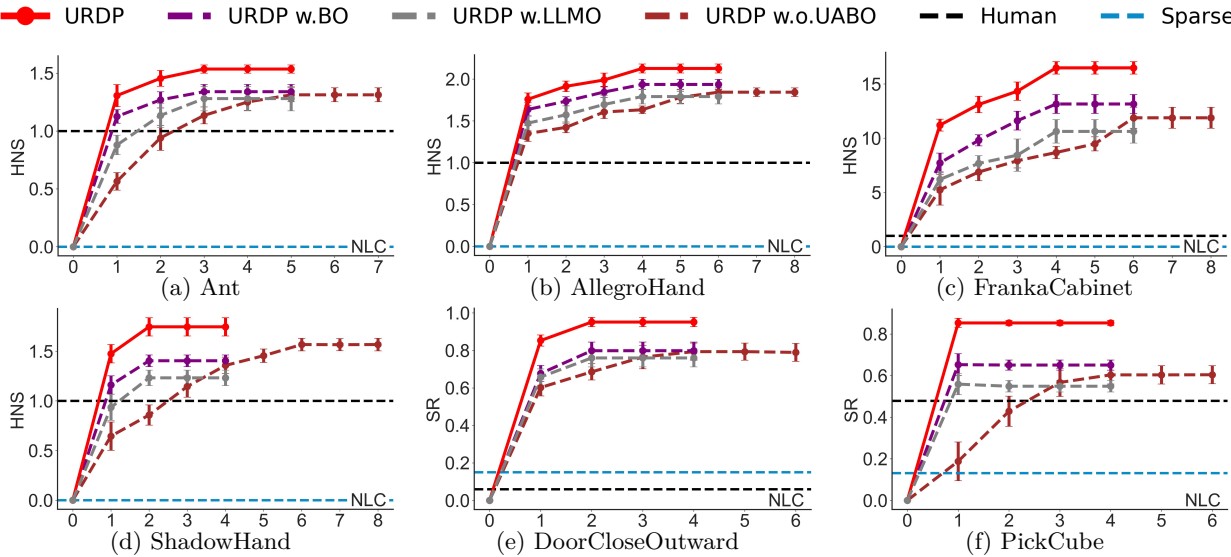

Figure 4: A comparison of the *URDP*, URDP w. BO, URDP w. LLMO and URDP w.o. UABO when all methods utilize identical simulation budgets (NOE).

**Abl-3: The UABO plays a key role in performance improvement**. This experimental study systematically compares the numerical optimization capabilities between large language models (LLMs) and Bayesian optimization approaches within our framework. In the first ablation, we replaced the UABO module with the LLM reflection (denoted as **URDP w. LLMO**), employing identical prompting strategies to Eureka, thereby configuring both outer-loop (reward components) and inner-loop (reward intensities) optimization entirely through LLMs. Figure 4 demonstrates that even under decoupled optimization conditions, LLM-based numerical optimization underperforms the Bayesian optimization, revealing fundamental limitations in mathematical optimization capabilities while confirming the critical role of Bayesian methods in enhancing the reward performance. These results substantiate our core hypothesis that LLMs serve more effectively as controllers for numerical optimization tools rather than direct optimizers. Subsequent validation experiments replacing UABO with standard BO (**URDP w. BO**) reveal UABO's superior efficiency (see Figure 5): *URDP* (UABO) achieves comparable or better performance than URDP w. BO using only 80% of the sampling budget across all Isaac tasks, with consistently superior final reward function performance, demonstrating that uncertainty-aware priors accelerate optimal search in the reward function design.

## 5.5 Extended Discussion

**Disc-1: Decoupling and performance degradation in evolutionary search.** As illustrated in Figure 6 (blue line), our experiments reveal performance degradation during evolutionary search in the baseline (Eureka) approach, with performance regression observed in 23% of Isaac tasks. Analysis of the LLMs'

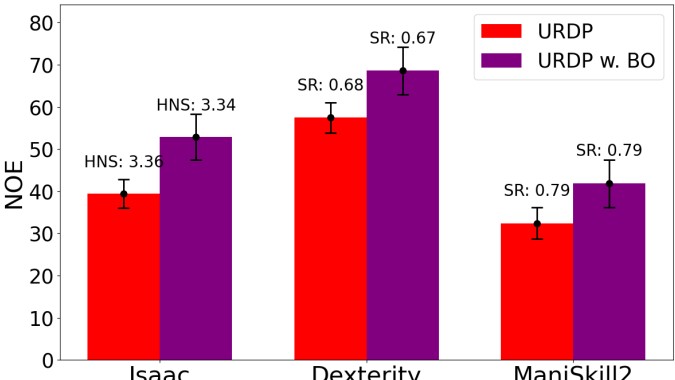

Figure 5: *URDP* achieves a significantly greater improvement in efficiency across all benchmarks compared to BO.

decision-making in these cases demonstrates that in 75% of instances, the models modified only the reward intensity hyperparameters while leaving the reward components unchanged, indicating a propensity for erroneous judgments in numerical optimization. More critically, we identify *oscillatory phenomenon* in the baseline optimization process (see Figure 6c), where LLMs entered persistent cycles of alternating between limited sets of hyperparameters during evolutionary search. This optimization instability resulted in complete convergence failure, with the baseline system trapped in ineffective, non-progressive iterations.

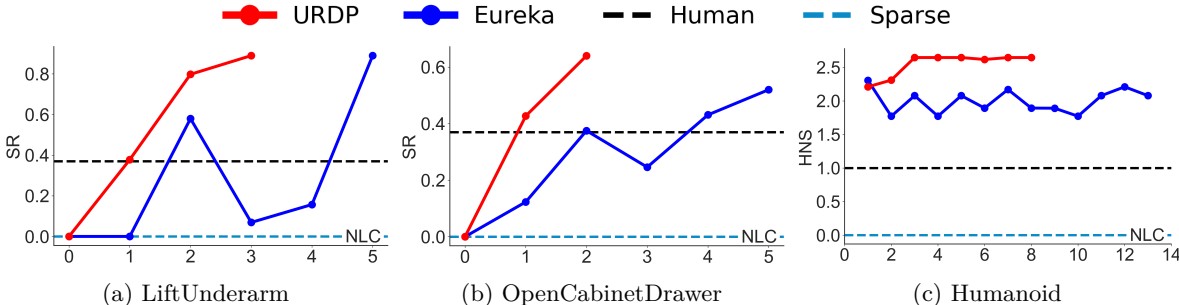

Figure 6: The baseline (Eureka) exhibits undesirable performance degradation during evolutionary search on certain tasks (a-b). Notably, the oscillatory phenomenon is detected in the baseline method for Task (c), indicating substantial computational waste of the baseline method.

In contrast, *URDP* demonstrates superior optimization efficiency, requiring significantly fewer evolutionary iterations (NLC) while exhibiting markedly reduced instances of performance regression and oscillatory behavior during the optimization process. Our empirical results show performance regression in merely 1% of Isaac tasks, with no observed cases of persistent optimization oscillations. These findings provide a strong empirical explanation for the effectiveness of the decoupled reward design approach implemented in *URDP*.

**Disc-2: The correlation between reward component uncertainty and novel reward discovery**. Our investigation reveals a noteworthy phenomenon: the high-uncertainty reward components ($r_{u\uparrow}$) identified by *URDP* frequently correspond to novel reward components not previously utilized in human-designed reward functions, with these components exhibiting significant reward shaping effects during RL training, thereby revealing a dual role of the uncertainty in enhancing both optimization efficiency and final policy performance. Here, we refer to a reward component as $r_{u\uparrow}$ when the uncertainty of the reward is greater than 0.9. Through systematic comparisons between reward components designed by the *URDP* and conventional human-designed rewards, Figure 7a illustrates a strong correlation between the component uncertainty level and the novelty, suggesting that the higher-uncertainty components may represent more innovative reward formulations. Ablation studies conducted by removing $r_{u\uparrow}$ components from the reward function ($R$ w.o. $r_{u\uparrow}$) and retraining

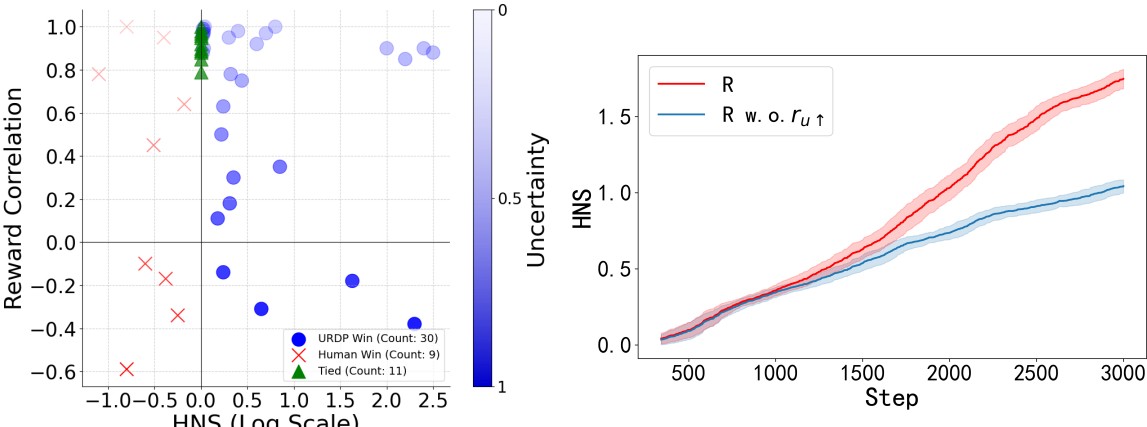

(a) *URDP* Rewards vs. Human Rewards on IsaacGym.

(b) RL training curves of reward functions R vs. R w.o. $r_{u\uparrow}$ on the ShadowHand.

Figure 7: (a) Results on IsaacGym tasks. Each marker represents one reward component. The $x$-axis shows the HNS on a log scale, while the $y$-axis gives the Pearson correlation between generated reward and human-designed reward. Marker color shows the component-level uncertainty $U_r$ (darker indicates higher uncertainty). Marker shapes indicate outcome relative to the human baseline (blue circles: URDP wins; red crosses: human wins; green triangles: ties). Tasks in the lower-right region illustrate reward formulations that deviate from human design yet achieve superior performance, suggesting that high-uncertainty components capture novel, previously unexplored reward. These components are `angvel_penalty` and `distance_penalty` in `ShadowHand`, `dof_vel_penalty` in `Anymal`, and `progress_reward` in `FrankaCabinet`. (b) ShadowHand training curves show that adding the higher-uncertainty component $r_{u\uparrow}$(angvel_penalty) to $R$ consistently yields higher HNS than the ablation without it.

PPO agents reveal statistically significant performance degradation and poorer convergence characteristics in the resulting policies as shown in Figure 7b, whereas the complete reward function maintains substantially better optimization stability, collectively providing conclusive evidence that $r_{u\uparrow}$ components play an essential role in effective reward shaping and policy learning regularization. See more examples in App. G.1.

# 6 Conclusion

In this work, we present a novel decoupled architecture for automated reward function design in reinforcement learning. Our framework introduces uncertainty quantification into reward component design, significantly improving the sampling efficiency of LLMs. Furthermore, we propose Uncertainty-Aware Bayesian Optimization to enable efficient hyperparameter search. Extensive experimental results demonstrate that our approach outperforms existing methods in both reward function quality and automated design efficiency.

## Acknowledgments

This work has been supported by the program of the National Natural Science Foundation of China (No. 61906195) and the independent deployment project of the National Key Laboratory of Cognition and Decision Intelligence for Complex Systems, Institution of Automation, Chinese Academy of Sciences.

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

## A  Full Prompts

In this section, we provide all prompts and a concrete example of the feedback used in the *URDP* framework.

Prompt 1: Initial system prompt

```
You are a reward engineer trying to write reward functions to solve reinforcement learning tasks as effective as possible.
Your goal is to write a reward function for the environment that will help the agent learn the task described in text.
Your reward function should use useful variables from the environment as inputs. As an example,
the reward function signature can be:
@torch.jit.script
def compute_reward(object_pos: torch.Tensor, goal_pos: torch.Tensor) -> Tuple[torch.Tensor, Dict[str, torch.Tensor]]:
    ...
    return reward, {}
Since the reward function will be decorated with @torch.jit.script,
please make sure that the code is compatible with TorchScript (e.g., use torch tensor instead of numpy array).
Make sure any new tensor or variable you introduce is on the same device as the input tensors.
```

Prompt 2: Reward reflection and feedback

```
We trained a RL policy using the provided reward function code and tracked the values of the individual components in the
reward function as well as global policy metrics such as success rates and episode lengths after every {epoch_freq} epochs
and the maximum, mean, minimum values encountered:
<REWARD REFLECTION HERE1>

We calculated a score for each sample based on the uncertainty of the reward term. We then calculated the standard and
extreme deviations of all the sample scores in this iteration, which were as follows:
<REWARD REFLECTION HERE2>

Please adopt the following recommendations for the next iteration of reward function generation:
    (1) If the standard deviation is less than 0.05 and the extreme deviation is less than 0.1, it is recommended to stop the
    iteration and instead encourage exploration of new combinations of reward terms.
    (2) It is recommended to retain reward items with higher scores in the sample and remove those with lower scores.
    (3) Only the combination of reward items and the content of the reward function need to be optimised, not the numerical
    optimisation.

Please carefully analyze the policy feedback and provide a new, improved reward function that can better solve the task.
Some helpful tips for analyzing the policy feedback:
    (1) If the success rates are always near zero, then you must rewrite the entire reward function
    (2) If some reward components' magnitude is significantly larger, then you must re-scale its value to a proper range
Please analyze each existing reward component in the suggested manner above first, and then write the reward function code.
```

Prompt 3: Code formatting tip

```
The output of the reward function should consist of two items:
    (1) the total reward,
    (2) a dictionary of each individual reward component.
The code output should be formatted as a python code string: "```python ... ```".

Some helpful tips for writing the reward function code:
    (1) You may find it helpful to normalize the reward to a fixed range by applying transformations like torch.exp to the
    overall reward or its components
    (2) If you choose to transform a reward component, then you must also introduce a temperature parameter inside the
    transformation function; this parameter must be a named variable in the reward function and it must not be an input
    variable. Each transformed reward component should have its own temperature variable
    (3) Make sure the type of each input variable is correctly specified; a float input variable should not be specified as
    torch.Tensor
    (4) Most importantly, the reward code's input variables must contain only attributes of the provided environment class
    definition (namely, variables that have prefix self.). Under no circumstance can you introduce new input variables.
```

Example of feedback

```
Iteration 1: URDP Best Sample:
```python
@torch.jit.script
def compute_reward(
    object_rot: torch.Tensor,
    goal_rot: torch.Tensor,
    object_angvel: torch.Tensor,
```

```
        fingertip_pos: torch.Tensor,
        object_pos: torch.Tensor,
        actions: torch.Tensor
) -> Tuple[torch.Tensor, Dict[str, torch.Tensor]]:
    """
    Reward function for spinning an object to target orientation.

    orientation_temp: float = 1.0
    alignment_temp: float = 0.5
    spin_temp: float = 0.3
    control_penalty_temp: float = -0.01
    fingertip_dist_temp: float = -0.1

    device = object_rot.device

    # 1. Orientation alignment reward (main objective)
    # Quaternion dot product (cosine of half the angle between rotations)
    quat_diff = quat_mul(object_rot, quat_conjugate(goal_rot))
    orientation_reward = orientation_temp * (1.0 - torch.abs(quat_diff[:, 0]))

    # 2. Angular velocity alignment with the desired rotation axis
    # We want the object to be spinning, not just static at target orientation
    avg_angvel = torch.mean(torch.abs(object_angvel), dim=1)
    spin_reward = spin_temp * avg_angvel

    # 3. Control penalty (small penalty for large actions)
    action_penalty = control_penalty_temp * torch.mean(torch.square(actions), dim=1)

    # 4. Fingertip distance penalty (keep fingertips close to object)
    fingertip_dist = torch.norm(fingertip_pos - object_pos.unsqueeze(1), dim=-1)
    avg_fingertip_dist = torch.mean(fingertip_dist, dim=1)
    distance_penalty = fingertip_dist_temp * avg_fingertip_dist

    # Combine all components
    total_reward = orientation_reward + spin_reward + action_penalty + distance_penalty

    # Return components for debugging
    reward_dict = {
        "orientation_reward": orientation_reward,
        "spin_reward": spin_reward,
        "action_penalty": action_penalty,
        "distance_penalty": distance_penalty
    }

    return total_reward, reward_dict
```

Iteration 1: Feedback:
We trained an RL policy using the provided reward function code and tracked the values of the individual components in the reward function, as well as global policy metrics such as success rates and episode lengths after every 30 epochs, and the maximum, mean, and minimum values encountered:

Best Sample (0) metrics:
orientation_reward: ['0.57', '0.60', '0.61', '0.58', '0.57', '0.59', '0.58', '0.58', '0.57', '0.57'], Max: 0.61, Mean: 0.58, Min: 0.56
spin_reward: ['2.10', '2.12', '2.28', '2.42', '2.78', '2.84', '3.07', '4.01', '4.74', '5.09'], Max: 5.62, Mean: 3.31, Min: 1.79
action_penalty: ['-0.19', '-0.21', '-0.22', '-0.23', '-0.24', '-0.25', '-0.26', '-0.27', '-0.28', '-0.29'], Max: -0.19, Mean: -0.25, Min: -0.30
distance_penalty: ['-0.03', '-0.03', '-0.03', '-0.03', '-0.03', '-0.03', '-0.03', '-0.02', '-0.02', '-0.02'], Max: -0.02, Mean: -0.03, Min: -0.03
success_rate: ['0.00', '0.00', '0.05', '0.01', '0.03', '0.01', '0.08', '0.04', '0.02', '0.02'], Max: 0.20, Mean: 0.03, Min: 0.00
episode_lengths: ['6.88', '85.14', '177.95', '218.96', '263.25', '298.06', '308.33', '311.40', '305.05', '322.21'], Max: 326.63, Mean: 243.48, Min: 6.88

We calculated a score for each sample based on the uncertainty of the reward term. We then calculated the standard and extreme deviations of all the sample scores in this iteration, which were as follows:

Sample Score Statistics:
Standard Deviation: 0.080
Range: 0.309
Sample 12 metrics:
orientation_reward: ['0.07', '0.06', '0.07', '0.07', '0.07', '0.07', '0.07', '0.07', '0.07', '0.06'], Max: 0.07, Mean: 0.07, Min: 0.06
angvel_reward: ['0.00', '0.01', '0.00', '0.01', '0.00', '0.01', '0.01', '0.01', '0.01', '0.01'], Max: 0.01, Min: 0.00
dof_vel_penalty: ['68.31', '74.19', '74.67', '80.21', '86.64', '93.44', '98.93', '103.92', '105.69', '107.19'], Max: 109.81, Mean: 91.46, Min: 67.74
fingertip_distance_penalty: ['0.07', '0.09', '0.08', '0.07', '0.08', '0.08', '0.07', '0.07', '0.07', '0.08'], Max: 0.09, Mean: 0.08, Min: 0.07

```
success_rate: ['0.00', '0.03', '0.01', '0.00', '0.02', '0.01', '0.02', '0.00', '0.00', '0.00'], Max: 0.18, Mean: 0.02, Min:
0.00
episode_lengths: ['6.88', '112.69', '217.70', '263.16', '267.88', '352.34', '346.89', '360.60', '348.12', '355.96'], Max:
379.91, Mean: 275.82, Min: 6.88

Sample 6 metrics:
orientation_reward: ['0.66', '0.65', '0.65', '0.65', '0.65', '0.66', '0.66', '0.65', '0.66', '0.66'], Max: 0.66, Mean: 0.66,
Min: 0.65
angvel_reward: ['0.30', '0.43', '0.53', '0.53', '0.51', '0.47', '0.49', '0.49', '0.52', '0.50'], Max: 0.53, Mean: 0.48, Min:
0.27
stability_reward: ['0.96', '0.97', '0.97', '0.98', '0.98', '0.98', '0.98', '0.98', '0.98', '0.98'], Max: 0.98, Mean: 0.98,
Min: 0.95
contact_reward: ['0.99', '0.99', '0.99', '0.99', '0.99', '0.99', '0.99', '0.99', '0.99', '0.99'], Max: 0.99, Mean: 0.99, Min:
0.99
action_reg: ['-0.05', '-0.06', '-0.06', '-0.07', '-0.07', '-0.07', '-0.08', '-0.08', '-0.08', '-0.09'], Max: -0.05, Mean:
-0.07, Min: -0.09
success_rate: ['0.00', '0.02', '0.00', '0.04', '0.01', '0.00', '0.01', '0.01', '0.04', '0.00'], Max: 0.13, Mean: 0.02, Min:
0.00
episode_lengths: ['6.88', '104.14', '264.72', '234.71', '321.16', '351.02', '347.56', '327.18', '343.47', '341.25'], Max:
395.64, Mean: 272.77, Min: 6.88

Sample 8 metrics:
orientation_reward: ['0.35', '0.35', '0.35', '0.35', '0.35', '0.35', '0.35', '0.35', '0.35', '0.35'], Max: 0.36, Mean: 0.35,
Min: 0.34
angvel_reward: ['0.36', '0.62', '0.68', '0.75', '0.74', '0.71', '0.79', '0.78', '0.84', '0.85'], Max: 0.88, Mean: 0.74, Min:
0.35
position_reward: ['1.00', '1.00', '1.00', '1.00', '1.00', '0.99', '0.99', '0.99', '0.99', '0.99'], Max: 1.00, Mean: 1.00,
Min: 0.99
contact_reward: ['1.00', '1.00', '1.00', '1.00', '1.00', '1.00', '1.00', '1.00', '1.00', '1.00'], Max: 1.00, Mean: 1.00,
Min: 1.00
action_penalty: ['-0.10', '-0.12', '-0.14', '-0.14', '-0.15', '-0.15', '-0.15', '-0.15', '-0.15', '-0.16'], Max: -0.10,
Mean: -0.15, Min: -0.17
success_rate: ['0.00', '0.00', '0.01', '0.01', '0.02', '0.00', '0.01', '0.03', '0.02', '0.05'], Max: 0.14, Mean: 0.02, Min:
0.00
episode_lengths: ['6.88', '107.01', '227.35', '219.53', '249.13', '292.37', '281.23', '331.81', '298.09', '305.40'], Max:
399.00, Mean: 250.83, Min: 6.88

Sample 10 metrics:
orientation_reward: ['0.80', '0.80', '0.80', '0.80', '0.80', '0.80', '0.80', '0.80', '0.80', '0.80'], Max: 0.81, Mean: 0.80,
Min: 0.80
angvel_reward: ['0.56', '0.69', '0.80', '0.88', '0.92', '0.91', '0.93', '0.94', '0.94', '0.94'], Max: 0.96, Mean: 0.87, Min:
0.54
fingertip_reward: ['1.00', '1.00', '1.00', '1.00', '1.00', '1.00', '1.00', '1.00', '1.00', '1.00'], Max: 1.00, Mean: 1.00,
Min: 1.00
action_reward: ['0.99', '0.99', '0.99', '0.98', '0.98', '0.98', '0.98', '0.98', '0.98', '0.98'], Max: 0.99, Mean: 0.98, Min:
0.98
success_rate: ['0.00', '0.01', '0.02', '0.03', '0.02', '0.00', '0.01', '0.00', '0.00', '0.00'], Max: 0.17, Mean: 0.01, Min:
0.00
episode_lengths: ['6.88', '104.65', '285.08', '312.46', '354.28', '356.70', '275.36', '371.96', '368.43', '370.56'], Max:
395.92, Mean: 296.21, Min: 6.88

Sample 3 metrics:
orientation_reward: ['0.00', '0.00', '0.00', '0.00', '0.00', '0.00', '0.00', '0.00', '0.00', '0.00'], Max: 0.00, Mean: 0.00,
Min: 0.00
angvel_reward: ['0.36', '0.68', '0.81', '0.87', '0.85', '0.90', '0.89', '0.92', '0.92', '0.92'], Max: 0.93, Mean: 0.84, Min:
0.35
fingertip_reward: ['-0.00', '-0.00', '-0.00', '-0.00', '-0.00', '-0.00', '-0.00', '-0.00', '-0.00', '-0.00'], Max: -0.00,
Mean: -0.00, Min: -0.00
action_penalty: ['-0.01', '-0.01', '-0.02', '-0.02', '-0.02', '-0.02', '-0.02', '-0.02', '-0.02', '-0.02'], Max: -0.01,
Mean: -0.02, Min: -0.02
object_vel_penalty: ['-0.00', '-0.00', '-0.00', '-0.00', '-0.00', '-0.00', '-0.00', '-0.00', '-0.00', '-0.00'], Max: -0.00,
Mean: -0.00, Min: -0.00
success_rate: ['0.00', '0.02', '0.01', '0.00', '0.00', '0.00', '0.01', '0.01', '0.00', '0.00'], Max: 0.05, Mean: 0.01, Min:
0.00
episode_lengths: ['6.88', '83.31', '208.44', '169.25', '228.87', '225.52', '222.20', '236.94', '254.80', '282.27'], Max:
399.00, Mean: 209.30, Min: 6.88

Please adopt the following recommendations for the next iteration of reward function generation:
(1) If the standard deviation is less than 0.05 and the extreme deviation is less than 0.1, it is recommended to stop the
iteration and instead encourage exploration of new combinations of reward terms.
(2) For all reward items with higher scores it is recommended to keep them and for those with lower scores it is recommended
to remove them.
(3) Only the combination of reward items and the content of the reward function need to be optimised, not the numerical
optimisation.

Please carefully analyze the policy feedback and provide a new, improved reward function that can better solve the task.
Some helpful tips for analyzing the policy feedback:
    (1) If the success rates are always near zero, then you must rewrite the entire reward function
    (2) If some reward components' magnitude is significantly larger, then you must re-scale its value to a proper range
```

# B  Benchmark Details

## B.1  An Introduction to the Benchmarks

**Isaac.**The Isaac Gym benchmark includes a broad set of continuous control tasks, covering locomotion, balancing, aerial control, and dexterous manipulation. Robots range from low-DoF systems (e.g., Cartpole, Ball Balance) to complex agents such as Humanoid, Anymal, AllegroHand, and ShadowHand. Each task presents different control challenges, requiring precise joint coordination, stable gait generation, or fine-grained object interaction. All tasks provide observations, including joint positions, velocities, root orientation, and task-specific data such as object pose or goal location. The control mode varies by task—torque, velocity, or end-effector control—depending on the robot type. Randomization of initial states and physical parameters (e.g., mass, friction) is applied during training to improve robustness and generalization. The benchmark emphasizes both low-level motor control and high-level strategy in physics-rich environments.

**Dexterity.**The Dexterous benchmark focuses on dexterous manipulation using the 24-DoF ShadowHand across a wide range of object-centric tasks. These include stacking blocks, turning faucets, opening doors, rotating bottles, catching objects, and tool use. The tasks require precise finger control, contact-rich interactions, and adaptability to diverse object geometries and behaviors. Each environment provides proprioceptive input (joint states, fingertip positions), as well as object-related observations (pose, velocity, goal state). Control is applied via joint position or velocity commands. The tasks involve significant variability in object placement, orientation, and physical properties, encouraging the development of general and robust manipulation policies. The benchmark highlights the challenge of high-dimensional motor coordination in real-world-like, unstructured settings.

**Maniskill2**. In the ManiSkill2 environment, a 7-DoF Franka Panda robotic arm is used by default. For tasks focused on stationary manipulation—such as Lift Cube, Pick Cube, Turn Faucet, and Stack Cube—a fixed-base arm configuration is employed. In contrast, tasks involving mobility, such as Open Cabinet Door and Open Cabinet Drawer, utilize a single-arm robot mounted on a Sciurus17 mobile base. The Push Chair task is handled by a dual-arm system, also equipped with the Sciurus17 base. The observation space includes robot-centric data like joint angles, joint velocities, and the base's pose (position and orientation in the world frame), along with task-specific inputs such as goal coordinates and end-effector locations. Control is performed in end-effector delta pose mode, which directly manages changes in 3D translation and orientation, the latter expressed in axis-angle form relative to the end-effector's frame. Each task features variability in key parameters, including the initial and goal states of the object being manipulated, the robot's starting joint configuration, and physical dynamics like friction and damping.

## B.2  ManiSkill2 Environment Details

We provide detailed information about the ManiSkill2 environment in this section. Detailed information about the Isaac and Dexterity environments is the same as in the Eureka (see the content in the appendix of the paper (Ma et al., 2024a)). For each environment, we list its observation and action dimensions, the original description of the task, and the task fitness function $F$.

| ManiSkill2 Environments |
|---|
| Environment (obs dim, action dim) 
 Task description 
 Task fitness function $F$ |
| **PickCube-v0** (51,7) 
 This class corresponds to the PickCube task in ManiSkill. This environment consists of a robot arm and a cube placed on the table. At the beginning, the cube appears at a random location and orientation. The agent must control the gripper to approach, grasp, and lift the cube above a threshold height. The challenge lies in object localization, precise control, and stable grasping. 
 $1[\text{dist\_cube\_goal} < 0.05]$ |
| **LiftCube-v0** (42,7) 
 This environment corresponds to the LiftCube task. The agent is required to grasp a cube and lift it vertically above a specific height threshold. The task emphasizes accurate vertical movement and stable grasping without disturbing the cube's pose. 
 $1[\text{cube\_height} > 0.2]$ |
| **TurnFaucet-v0** (40, 7) 
 This class corresponds to the TurnFaucet task. A faucet handle is mounted on a wall, and the agent must rotate it clockwise or counterclockwise to a target angle. The challenge lies in establishing proper contact, applying sufficient torque, and maintaining stability during the turning motion. 
 $1[\text{rotation\_reward} < 0.1]$ |
| **OpenCabinetDoor-v1** (75, 11) 
 This environment corresponds to the OpenCabinetDoor task. A cabinet with a side-hinged door is presented. The agent must locate and pull the door handle to open it. The task involves estimating the door's hinge axis, approaching from an appropriate angle, and applying a pulling force that aligns with the door's rotation. 
 $1[\text{goal\_diff} < 0.1 \text{ and is\_static}]$ |
| **OpenCabinetDrawer-v1** (75, 11) 
 This class corresponds to the OpenCabinetDrawer task. The robot must open a drawer embedded in a cabinet by locating the handle and pulling it outward. The task requires both accurate handle grasping and force application along a linear trajectory, while avoiding excessive torque that could misalign the drawer. 
 $1[\text{goal\_diff} < 0.05 \text{ and is\_static}]$ |
| **PushChair-v1** (131, 18) 
 This environment corresponds to the PushChair task. The robot must push a movable chair from its initial location to a designated target region. The chair is free to rotate and slide. The agent needs to make strategic contact with the chair body and adjust its pushing direction dynamically to avoid misalignment and ensure accurate placement. 
 $1[\text{chair\_to\_target\_dist} < 0.3 \text{ and chair\_tilt} < 0.2]$ |

## C  Implementation Details

### C.1  Implementation Details of Sampling and Uncertainty

When explicitly prompted to generate diverse outputs, LLMs inevitably produce varying textual expressions for semantically equivalent content - a phenomenon particularly evident in reward function generation where code implementations may differ lexically while encoding identical reward semantics. For instance, as demonstrated in Section D.2, two LLM-generated reward function samples might both incorporate velocity-based rewards while exhibiting completely different textual formulations. Failure to detect and eliminate such semantic redundancies leads to computationally expensive duplicate evaluations that cannot be effectively identified through surface-level text matching, necessitating deeper semantic analysis for accurate deduplication.

Therefore, the `URDP` utilizes the BGE-M3 model (Xiao et al., 2024) for the purpose of semantic similarity assessment, whereas the built-in SequenceMatcher in Python is employed for text similarity assessment. The uncertainty quantification for both reward components and reward functions is implemented through similarity comparison. Specifically, the component uncertainty score ($U_{r_{i,:}}$) is computed by comparing a given reward component against all components generated within the same iteration. The reward function uncertainty ($U_{R_i}$) score is derived through comparison with all functionally similar reward functions from the same iteration (referred to as a similarity group). From each similarity group, only one reward function is randomly selected for training, while the remaining ones are discarded, thereby filtering out redundant reward functions. See Alg. 2 for the pseudocode. We employ a similarity threshold of 0.95, where the final similarity metric is determined as the maximum value between semantic similarity and textual similarity.

---

**Algorithm 2:** Uncertainty Quantification in the `URDP`

---

**Input:** $K$ reward component samples $\{r_{i,1}, r_{i,2}, \ldots, r_{i,m}\}_{i \in K}$, text models $S_{text}$ and semantic models $S_{semantic}$.

**Output:** Reward components uncertainty $\{U_{r_{i,1}}, \ldots, U_{r_{i,m}}\}$, reward functions uncertainty $U_{R_i}$ and similarity sample group $B_i$.

**foreach** $R_i \in K$ **do**

   **foreach** $r_{i,1}, r_{i,2}, \ldots, r_{i,m}$ **do**

      **foreach** $r_{j,1}, r_{j,2}, \ldots, r_{j,m}$ $with\ j > i$ **do**

         **if** $\max(S_{text}(r_{i,m}, r_{j,m}),\ S_{semantic}(r_{i,m}, r_{j,m})) > 0.95$ **then**

            $\text{count}_m \leftarrow \text{count}_m + 1$

      $U_{r_{i,m}} \leftarrow 1 - \text{count}_m/K$, $U_{r_i} \leftarrow \sum_{m=1}^{M_i} U_{r_{i,m}}$

   $U_{R_i} \leftarrow U_{r_i}/(U_{r_i} + \cdots + U_{r_k})$

   **foreach** $R_{j>i}$ **do**

      **if** $\max(S_{text}(R_i, R_j),\ S_{semantic}(R_i, R_j)) > 0.95$ ***and*** $R_i$ *not in other* $B$ **then**

         Add $R_j$ to $B_i$

**return** $\{U_{r_{i,1}}, \ldots, U_{r_{i,m}}\}$, $U_{R_i}$, $B_{i,n=1}$

---

## C.2 Hyper-parameter Settings

All hyperparameters in `URDP` are listed in Table 3. The reinforcement learning algorithms employed for validation maintain the default configurations specified for each respective environment, with all hyperparameters comprehensively documented in Tables 4 and 5.

Table 3: Hyperparameters of `URDP`.

| Hyper-parameter | Value |
|---|---|
| Quantity of the reward samples $K$ | 16 |
| Maximum # of iterations $N_{outer}$ | 10 |
| Baseline # of iterations $N_{inner}$ | 10 |
| Maximum similarity $\omega$ | 0.95 |
| Smoothness $\nu$ | 2.5 |
| Length scale $\ell$ | $U_R$ |

We set the quantity of the reward samples $K$ to be the same as that used in Eureka, ensuring fairness in comparisons.

**Determination of outer maximum iterations $N_{outer}$ and inner baseline iterations $N_{inner}$.** The settings for these two hyperparameters were determined based on experimentally observed safe upper bounds. For the outer maximum iterations $N_{outer}$, Figure 8a compares the average NLC (and its variation range, indicated by black lines) required by `URDP` and Eureka across different benchmarks when achieving comparable

HNS and SR. Since the variation range of NLC is within 10 for all cases, we set the maximum iterations $N_{outer} = 10$ as a safe upper bound, ensuring effective completion of all tasks. For the inner baseline iterations $N_{inner}$, Figure 8b(b) presents the average number of inner-loop iterations required by `URDP` across the three benchmarks, along with their ranges. The required range is also within 10, while in Alg. 1 the inner-loop iteration count is defined as $N_{inner} \cdot U_R$ (with $U_R \in [0, 1]$). Therefore, we set $N_{inner} = 10$ to meet the requirements across tasks.

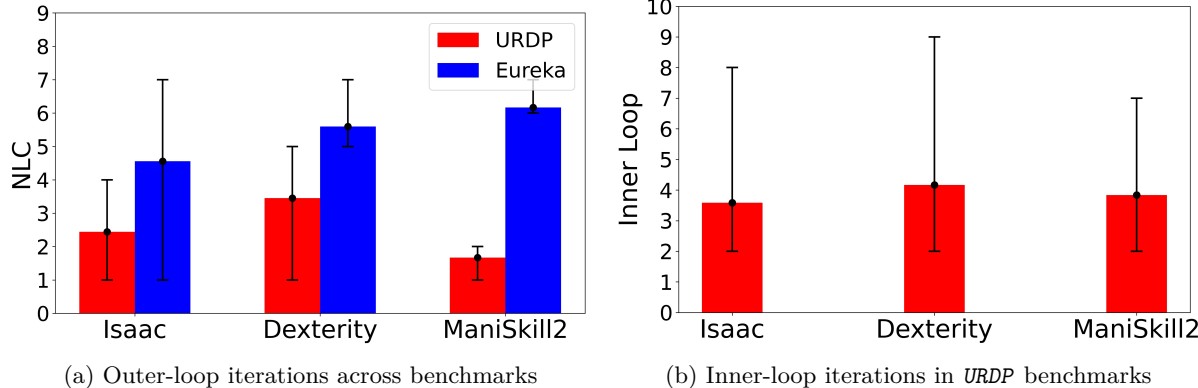

(a) Outer-loop iterations across benchmarks  (b) Inner-loop iterations in `URDP` benchmarks

Figure 8: Statistics of the outer-loop (equivalent to NLC) and inner-loop iteration counts. (a) The number of outer-loop iterations required across different benchmarks, with black lines indicating the variation range. (b) The statistics and variation ranges of inner-loop iterations for `URDP` during the experiments.

**Influence of maximum similarity** $\omega$**.** The maximum similarity parameter $\omega$ primarily determines the criterion for assessing sample similarity. Both the reward function and the uncertainty scores $U_R$ and $U_r$ designed in `URDP` are influenced by this parameter, thereby affecting the overall task score. To examine its impact, we conducted an ablation study by setting $\omega$ to 0.9 (denoted as `URDP`$_{\omega=0.9}$) and compared it with our default setting of $\omega = 0.95$. As shown in Figure 9a, reducing $\omega$ to 0.9 leads to a decline in both the success rate during the training iterations and the final success rate. The main reason, illustrated in Figure 9b, is that with $\omega = 0.9$, fewer samples are evaluated per iteration. This occurs because the more lenient similarity threshold causes `URDP` to overlook some high-quality samples, ultimately resulting in a drop in success scores.

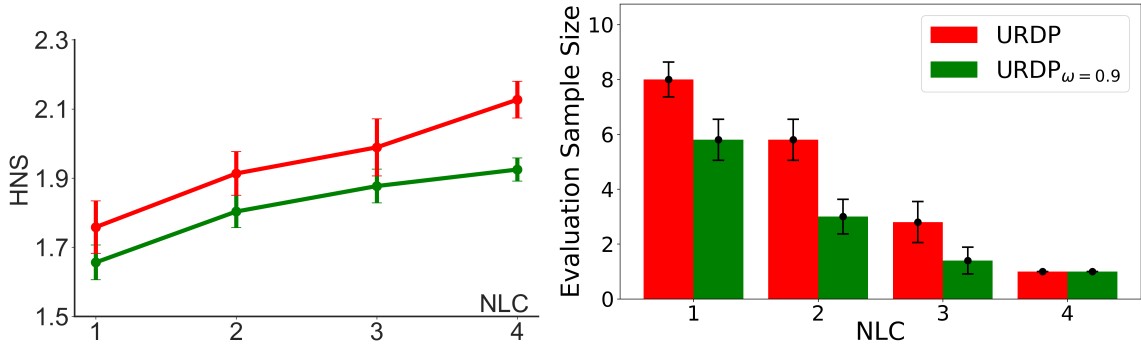

(a) `URDP` outperforms `URDP`$_{\omega=0.9}$ in the Allergo-  (b) The number of samples to be evaluated in the NLC
Hand task.  process on the AllegroHand task.

Figure 9: Results of the AllegroHand task under `URDP` and `URDP`$_{\omega=0.9}$. (a) HNS variation during the NLC process, where `URDP`$_{\omega=0.9}$ remains consistently lower than `URDP`. (b) Reducing $\omega$ in NLC substantially decreases the number of evaluated samples.

**Effect of Smoothness Parameter** $\nu$**.** The parameter $\nu$ determines the differentiability of the kernel function, which directly influences the trade-off between exploration and exploitation during Bayesian optimization. In theory, smaller $\nu$ values correspond to less smooth kernels that encourage more aggressive exploration. To

evaluate the effect of the Matern kernel smoothness parameter $\nu = 2.5$ in UABO on *URDP*, we conduct an ablation study by setting different commonly used $\nu$ values (0.5 and 1.5, denoted as **URDP w. UABO$_{0.5}$** and **URDP w. UABO$_{1.5}$**, respectively). As shown in Figure 10, when achieving comparable or identical HNS and SR, **URDP (UABO$_{2.5}$)** requires fewer NOE, indicating that the smoothness parameter $\nu$ not only enhances the efficiency of *URDP* but also aligns well with its theoretical motivation.

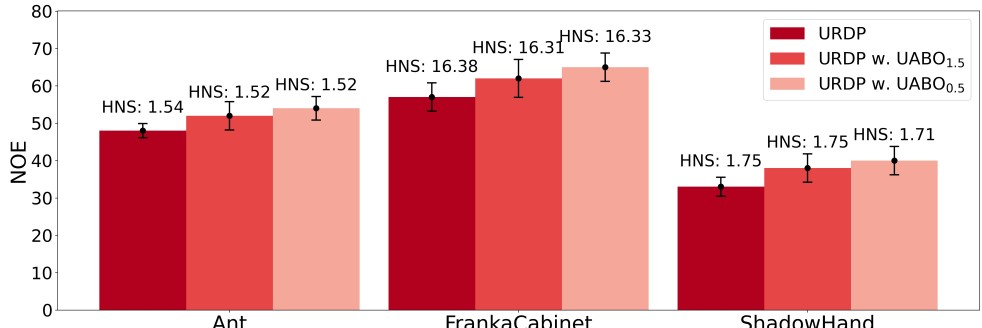

Figure 10: *URDP* with $\nu = 2.5$ outperforms other ablations in terms of efficiency.

**The impact of length scale $\ell$ on efficiency.** In the Matern kernel of a Gaussian Process, the length scale $\ell$ determines the rate at which the kernel value decays with distance, thereby affecting the exploration–exploitation balance of UABO in the parameter space. A larger $\ell$ slows the decay, increasing the correlation between distant sample points and encouraging global search, but may lead to slower convergence. In contrast, a smaller $\ell$ accelerates the decay, enhancing the sensitivity to local neighborhoods and favoring fine-grained local search, although at the risk of premature convergence. To balance these effects, *URDP* dynamically sets $\ell$ to the uncertainty score of the reward function $U_R$, whose value is in the range [0,1]. To assess the advantage of this adaptive length scale, we performed an ablation study using a fixed $\ell = 1$ (denoted *URDP* w. UABO$_1$) as a baseline, as shown in Figure 11. The experimental results show that $\ell = U_R$ (*URDP*) achieves comparable solution quality with fewer NOE, thus improving task efficiency.

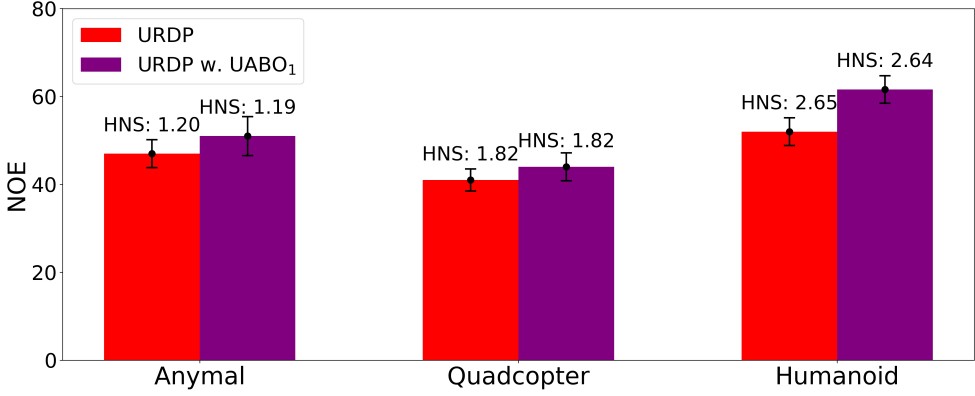

Figure 11: *URDP* outperforms *URDP* with $\ell = 1$(*URDP* w. UABO$_1$) in terms of efficiency.

# D Case Studies

## D.1 Case Study 1: LLMs in Numerical Optimization

This study employs two comparative examples to visualize the differences between *URDP* and Eureka in optimization processes. Example 1(a) and 1(b) present the respective design trajectories of *URDP* and Eureka for the ShadowHand task, where red annotations denote reward components and blue text indicates reward intensity hyperparameters.

Table 4: Hyperparameters of the SAC algorithm applied to Maniskill2.

| Hyper-parameter | Value |
|---|---|
| Discount factor $\gamma$ | 0.95 |
| Target update frequency | 1 |
| Learning rate | $3 \times 10^{-4}$ |
| Train frequency | 8 |
| Soft update $\tau$ | $5 \times 10^{-3}$ |
| Gradient steps | 4 |
| Learning starts | 4000 |
| Hidden units per layer | 256 |
| Batch Size | 1024 |
| # of layers | 2 |
| Initial temperature | 0.2 |
| Rollout steps per episode | 200 |

Table 5: Hyperparameters of the PPO algorithm applied to each task.

| Hyper-parameter | Value |
|---|---|
| Discount factor $\gamma$ | 0.99 (Isaac), 0.96 (Dexterity), 0.85 (ManiSkill2) |
| # of epochs per update | 8 (Isaac), 5 (Dexterity), 15 (ManiSkill2) |
| Learning rate | $5 \times 10^{-4}$ (Isaac), $3 \times 10^{-4}$ (Dexterity, ManiSkill2) |
| Batch size | 32768, 16384, 8192 (Isaac), 16384 (Dexterity), 400 (ManiSkill2) |
| Target KL divergence | 0.008 (Isaac), 0.016 (Dexterity), 0.05 (ManiSkill2) |
| # of layers | 3 (Isaac, Dexterity), 2 (ManiSkill2) |
| # of steps per update | 16 (Isaac), 8 (Dexterity), 3200 (ManiSkill2) |

Analysis of Example 1(a) demonstrates that during evolutionary search iterations, Eureka exclusively modifies reward intensity hyperparameters while failing to improve reward components. Despite multiple optimization attempts, this approach yields degraded performance. This finding reveals a critical limitation: when simultaneously optimizing both reward components and their strengths, LLMs cannot effectively utilize their inherent advantages in semantic correlation and autoregressive modeling, while their deficiencies in numerical optimization become particularly pronounced.

In contrast, *URDP*'s decoupled alternating optimization demonstrates systematic effectiveness. As shown in Example 1(b), *URDP* proactively explores novel reward components (e.g., replacing "distance reward" with "velocity reward"), achieving a 70.3% performance improvement (from 0.929 to 1.583) before hyperparameter optimization. Subsequent Bayesian optimization further enhances performance with an average 22.1% gain, culminating in a 102.5% total improvement over the initial configuration. Example 2 confirms this consistent advantage, providing compelling visual evidence of *URDP*'s superior optimization characteristics.

Example 1(a): Eureka reward functions on ShadowHand (Iteration 2, HNS: 1.045; Iteration 3, HNS: 0.898; Iteration 4, HNS: 1.019; Iteration 5, HNS: 0.955).

```
Iteration 2:
def compute_reward(object_rot: torch.Tensor, goal_rot: torch.Tensor, object_angvel: torch.Tensor) -> Tuple[torch.Tensor,
Dict[str, torch.Tensor]]:
    # Temperature parameters for reward components
    orientation_temp: float = 5.0  # Increased sensitivity for orientation alignment
    angular_velocity_temp: float = 0.5  # Adjusted sensitivity for angular velocity

    # Compute the difference in orientation between the object and the goal
    orientation_diff = 1.0 - torch.abs(torch.sum(object_rot * goal_rot, dim=-1))  # Quaternion dot product

    # Reward for minimizing the orientation difference
    orientation_reward = torch.exp(-orientation_temp * orientation_diff)
```

```
12
13        # Reward for spinning the object (encouraging non-zero angular velocity)
14        angular_velocity_magnitude = torch.norm(object_angvel, dim=-1)
15        angular_velocity_reward = torch.tanh(angular_velocity_temp * angular_velocity_magnitude)  # Rescaled to [-1, 1]
16
17        # Combine the rewards with a focus on orientation alignment
18        total_reward = orientation_reward * (1.0 + angular_velocity_reward)  # Weighted combination
19
20        # Return the total reward and individual components
21        reward_components = {
22            "orientation_reward": orientation_reward,
23            "angular_velocity_reward": angular_velocity_reward
24        }
25        return total_reward, reward_components
26
27    ################################################################################
28
29    Iteration 3:
30    def compute_reward(object_rot: torch.Tensor, goal_rot: torch.Tensor, object_angvel: torch.Tensor) -> Tuple[torch.Tensor,
       Dict[str, torch.Tensor]]:
31        # Temperature parameters for reward components
32        orientation_temp: float = 10.0  # Increased sensitivity for orientation alignment
33        angular_velocity_temp: float = 1.0  # Adjusted sensitivity for angular velocity
34
35        # Compute the difference in orientation between the object and the goal
36        orientation_diff = 1.0 - torch.abs(torch.sum(object_rot * goal_rot, dim=-1))  # Quaternion dot product
37
38        # Reward for minimizing the orientation difference
39        orientation_reward = torch.exp(-orientation_temp * orientation_diff)
40
41        # Reward for spinning the object (encouraging non-zero angular velocity)
42        angular_velocity_magnitude = torch.norm(object_angvel, dim=-1)
43        angular_velocity_reward = torch.tanh(angular_velocity_temp * angular_velocity_magnitude)  # Rescaled to [-1, 1]
44
45        # Combine the rewards with a focus on orientation alignment
46        total_reward = orientation_reward * (1.0 + angular_velocity_reward)  # Weighted combination
47
48        # Return the total reward and individual components
49        reward_components = {
50            "orientation_reward": orientation_reward,
51            "angular_velocity_reward": angular_velocity_reward
52        }
53        return total_reward, reward_components
54
55    ################################################################################
56
57    Iteration 4:
58    def compute_reward(object_rot: torch.Tensor, goal_rot: torch.Tensor, object_angvel: torch.Tensor) -> Tuple[torch.Tensor,
       Dict[str, torch.Tensor]]:
59        # Temperature parameters for reward components
60        orientation_temp: float = 10.0  # Increased sensitivity for orientation alignment
61        angular_velocity_temp: float = 2.0  # Adjusted sensitivity for angular velocity
62
63        # Compute the difference in orientation between the object and the goal
64        orientation_diff = 1.0 - torch.abs(torch.sum(object_rot * goal_rot, dim=-1))  # Quaternion dot product
65
66        # Reward for minimizing the orientation difference
67        orientation_reward = torch.exp(-orientation_temp * orientation_diff)
68
69        # Reward for spinning the object (encouraging non-zero angular velocity)
70        angular_velocity_magnitude = torch.norm(object_angvel, dim=-1)
71        angular_velocity_reward = torch.tanh(angular_velocity_temp * angular_velocity_magnitude)  # Rescaled to [-1, 1]
72
73        # Combine the rewards with a focus on orientation alignment
74        total_reward = orientation_reward * (1.0 + angular_velocity_reward)  # Weighted combination
75
76        # Return the total reward and individual components
77        reward_components = {
78            "orientation_reward": orientation_reward,
79            "angular_velocity_reward": angular_velocity_reward
80        }
81        return total_reward, reward_components
82
83    ################################################################################
84
85    Iteration 5:
86    def compute_reward(object_rot: torch.Tensor, goal_rot: torch.Tensor, object_angvel: torch.Tensor) -> Tuple[torch.Tensor,
       Dict[str, torch.Tensor]]:
87        # Temperature parameters for reward components
88        orientation_temp: float = 50.0  # Increased sensitivity for orientation alignment
89        angular_velocity_temp: float = 2.0  # Adjusted sensitivity for angular velocity
```

```
90
91         # Compute the difference in orientation between the object and the goal
92         orientation_diff = 1.0 - torch.abs(torch.sum(object_rot * goal_rot, dim=-1))  # Quaternion dot product
93
94         # Reward for minimizing the orientation difference
95         orientation_reward = torch.exp(-orientation_temp * orientation_diff)
96
97         # Reward for spinning the object (encouraging non-zero angular velocity)
98         angular_velocity_magnitude = torch.norm(object_angvel, dim=-1)
99         angular_velocity_reward = torch.tanh(angular_velocity_temp * angular_velocity_magnitude)  # Rescaled to [-1, 1]
100
101         # Combine the rewards with a focus on orientation alignment
102         total_reward = orientation_reward * (1.0 + angular_velocity_reward)  # Weighted combination
103
104         # Return the total reward and individual components
105         reward_components = {
106             "orientation_reward": orientation_reward,
107             "angular_velocity_reward": angular_velocity_reward
108         }
109         return total_reward, reward_components
```

Example 1(b): *URDP* reward functions on ShadowHand. Iteration 1 (before UABO), HNS: 0.929; Iteration 1 (after UABO), HNS: 1.165; Iteration 2 (before UABO), HNS: 1.583; Iteration 2 (after UABO), HNS: 1.882.

```
1   Iteration 1(before UABO):
2   def compute_reward(object_rot: torch.Tensor, goal_rot: torch.Tensor) -> Tuple[torch.Tensor, Dict[str, torch.Tensor]]:
3       # Temperature parameters
4       orientation_temp = 1.0  # Reduced temperature for better sensitivity
5       distance_temp = 1.0     # Temperature for distance-based reward
6
7       # Compute the dot product between the object and goal quaternions
8       dot_product = torch.sum(object_rot * goal_rot, dim=1)
9
10      # Ensure the dot product is within the valid range [-1, 1]
11      dot_product = torch.clamp(dot_product, -1.0, 1.0)
12
13      # Compute the angle difference between the quaternions
14      angle_diff = torch.acos(2.0 * dot_product**2 - 1.0)
15
16      # Orientation reward: exponential transformation of the angle difference
17      orientation_reward = torch.exp(-orientation_temp * angle_diff)
18
19      # Distance-based reward: encourages reducing the angle difference
20      distance_reward = -angle_diff  # Negative because we want to minimize the difference
21
22      # Success bonus: reward for achieving the target orientation
23      success_threshold: float = 0.05  # Easier threshold for success
24      success_bonus = torch.where(angle_diff < success_threshold, 100.0, 0.0)  # Larger bonus
25
26      # Total reward: weighted sum of orientation reward, distance reward, and success bonus
27      total_reward = orientation_reward + distance_reward + success_bonus
28
29      # Dictionary of individual reward components
30      reward_components = {
31          "orientation_reward": orientation_reward,
32          "distance_reward": distance_reward,
33      }
34
35      return total_reward, reward_components
36
37  #################################################################################################
38
39  Iteration 1(after UABO):
40  def compute_reward(object_rot: torch.Tensor, goal_rot: torch.Tensor) -> Tuple[torch.Tensor, Dict[str, torch.Tensor]]:
41      # Temperature parameters
42      orientation_temp = 1.1134  # Reduced temperature for better sensitivity
43      distance_temp = 1.1134     # Temperature for distance-based reward
44
45      # Compute the dot product between the object and goal quaternions
46      dot_product = torch.sum(object_rot * goal_rot, dim=1)
47
48      # Ensure the dot product is within the valid range [-1, 1]
49      dot_product = torch.clamp(dot_product, -1.0, 1.0)
50
51      # Compute the angle difference between the quaternions
52      angle_diff = torch.acos(2.0 * dot_product**2 - 1.0)
53
54      # Orientation reward: exponential transformation of the angle difference
```

```
55          orientation_reward = torch.exp(-orientation_temp * angle_diff)
56
57          # Distance-based reward: encourages reducing the angle difference
58          distance_reward = -angle_diff  # Negative because we want to minimize the difference
59
60          # Success bonus: reward for achieving the target orientation
61          success_threshold: float = 0.05  # Easier threshold for success
62          success_bonus = torch.where(angle_diff < success_threshold, 100.0, 0.0)  # Larger bonus
63
64          # Total reward: weighted sum of orientation reward, distance reward, and success bonus
65          total_reward = orientation_reward + distance_reward + success_bonus
66
67          # Dictionary of individual reward components
68          reward_components = {
69              "orientation_reward": orientation_reward,
70              "distance_reward": distance_reward
71          }
72
73          return total_reward, reward_components
74
75  ##############################################################################################
76
77  Iteration 2(before UABO):
78  def compute_reward(object_rot: torch.Tensor, goal_rot: torch.Tensor, object_angvel: torch.Tensor) -> Tuple[torch.Tensor,
    Dict[str, torch.Tensor]]:
79          # Temperature parameters for reward components
80          orientation_temp = 1
81          velocity_temp = 0.1
82
83          # Compute the difference in orientation between the object and the goal
84          orientation_diff = torch.norm(object_rot - goal_rot, dim=-1)
85
86          # Compute the angular velocity magnitude of the object
87          angvel_magnitude = torch.norm(object_angvel, dim=-1)
88
89          # Reward for minimizing the orientation difference
90          orientation_reward = torch.exp(-orientation_temp * orientation_diff)
91
92          # Reward for maintaining a high angular velocity (encourages spinning)
93          velocity_reward = torch.exp(-velocity_temp * (1.0 / (angvel_magnitude + 1e-6)))
94
95          # Combine the rewards with appropriate weights
96          total_reward = 0.7 * orientation_reward + 0.3 * velocity_reward
97
98          # Dictionary of individual reward components for logging
99          reward_dict = {
100             "orientation_reward": orientation_reward,
101             "velocity_reward": velocity_reward
102         }
103
104         return total_reward, reward_dict
105
106 ##############################################################################################
107
108 Iteration 2(after UABO):
109 def compute_reward(object_rot: torch.Tensor, goal_rot: torch.Tensor, object_angvel: torch.Tensor) -> Tuple[torch.Tensor,
    Dict[str, torch.Tensor]]:
110         # Temperature parameters for reward components
111         orientation_temp = 1.2642
112         velocity_temp = 0.3145
113
114         # Compute the difference in orientation between the object and the goal
115         orientation_diff = torch.norm(object_rot - goal_rot, dim=-1)
116
117         # Compute the angular velocity magnitude of the object
118         angvel_magnitude = torch.norm(object_angvel, dim=-1)
119
120         # Reward for minimizing the orientation difference
121         orientation_reward = torch.exp(-orientation_temp * orientation_diff)
122
123         # Reward for maintaining a high angular velocity (encourages spinning)
124         velocity_reward = torch.exp(-velocity_temp * (1.0 / (angvel_magnitude + 1e-6)))
125
126         # Combine the rewards with appropriate weights
127         total_reward = 0.7 * orientation_reward + 0.3 * velocity_reward
128
129         # Dictionary of individual reward components for logging
130         reward_dict = {
131             "orientation_reward": orientation_reward,
132             "velocity_reward": velocity_reward
133         }
```

```
134
135          return total_reward, reward_dict
```

Example 2 (a): Eureka reward functions on Humanoid (Iteration 1, HNS: 2.273; Iteration 2, score: 0.412; Iteration 3, HNS: 0.032; Iteration 4, HNS: 0.127).

```
1    Iteration 1:
2    def compute_reward(root_states: torch.Tensor, dt: float) -> Tuple[torch.Tensor, Dict[str, torch.Tensor]]:
3        # Extract the velocity of the humanoid's torso (root_states[:, 7:10] contains the linear velocity)
4        velocity = root_states[:, 7:10]
5
6        # Compute the forward speed (we assume the humanoid is moving along the x-axis)
7        forward_speed = velocity[:, 0]
8
9        # Reward for forward speed (scaled to a smaller range)
10       speed_temp = 1.0  # Reduced temperature for better scaling
11       speed_reward = forward_speed * speed_temp
12
13       # Reward for consistency (encourage maintaining high speed)
14       consistency_temp = 1.0
15       consistency_reward = torch.exp(-consistency_temp * torch.abs(forward_speed - torch.mean(forward_speed)))
16
17       # Total reward combines speed reward and consistency reward
18       reward = speed_reward + consistency_reward
19
20       # Dictionary of individual reward components
21       reward_dict = {
22           "speed_reward": speed_reward,
23           "consistency_reward": consistency_reward,
24       }
25
26       return reward, reward_dict
27
28    ##############################################################################################
29
30    Iteration 2:
31    def compute_reward(root_states: torch.Tensor, dt: float) -> Tuple[torch.Tensor, Dict[str, torch.Tensor]]:
32        # Extract the velocity of the humanoid's torso (root_states[:, 7:10] contains the linear velocity)
33        velocity = root_states[:, 7:10]
34
35        # Compute the forward speed (we assume the humanoid is moving along the x-axis)
36        forward_speed = velocity[:, 0]
37
38        # Reward for forward speed (scaled to a smaller range)
39        speed_temp = 0.2  # Further reduced temperature for better scaling
40        speed_reward = forward_speed * speed_temp
41
42        # Reward for consistency (encourage maintaining high speed over time)
43        consistency_temp = 0.5  # Increased temperature for better scaling
44        consistency_reward = torch.exp(-consistency_temp * torch.abs(forward_speed - torch.mean(forward_speed)))
45
46        # Total reward combines speed reward and consistency reward
47        reward = speed_reward + consistency_reward
48
49        # Dictionary of individual reward components
50        reward_dict = {
51            "speed_reward": speed_reward,
52            "consistency_reward": consistency_reward,
53        }
54
55        return reward, reward_dict
56
57    ##############################################################################################
58
59    Iteration 3:
60    def compute_reward(root_states: torch.Tensor, dt: float) -> Tuple[torch.Tensor, Dict[str, torch.Tensor]]:
61        # Extract the velocity of the humanoid's torso (root_states[:, 7:10] contains the linear velocity)
62        velocity = root_states[:, 7:10]
63
64        # Compute the forward speed (we assume the humanoid is moving along the x-axis)
65        forward_speed = velocity[:, 0]
66
67        # Reward for forward speed (scaled to a smaller range)
68        speed_temp = 0.1  # Reduced temperature for better scaling
69        speed_reward = forward_speed * speed_temp
70
71        # Reward for consistency (encourage maintaining high speed)
72        consistency_temp = 0.5  # Increased temperature for better scaling
```

```
73          consistency_reward = torch.exp(-consistency_temp * torch.abs(forward_speed - torch.mean(forward_speed)))
74
75          # Total reward combines speed reward and consistency reward
76          reward = speed_reward + consistency_reward
77
78          # Dictionary of individual reward components
79          reward_dict = {
80              "speed_reward": speed_reward,
81              "consistency_reward": consistency_reward,
82          }
83
84          return reward, reward_dict
85
86   ################################################################################################
87
88   Iteration 4:
89   def compute_reward(root_states: torch.Tensor, dt: float) -> Tuple[torch.Tensor, Dict[str, torch.Tensor]]:
90          # Extract the velocity of the humanoid's torso (root_states[:, 7:10] contains the linear velocity)
91          velocity = root_states[:, 7:10]
92
93          # Compute the forward speed (we assume the humanoid is moving along the x-axis)
94          forward_speed = velocity[:, 0]
95
96          # Reward for forward speed (scaled to a smaller range)
97          speed_temp = 0.1  # Further reduced temperature for better scaling
98          speed_reward = forward_speed * speed_temp
99
100         # Reward for consistency (encourage maintaining high speed over time)
101         consistency_temp = 10  # Increased temperature for better scaling
102         consistency_reward = torch.exp(-consistency_temp * torch.abs(forward_speed - torch.mean(forward_speed)))
103
104         # Total reward combines speed reward and consistency reward
105         reward = speed_reward + consistency_reward
106
107         # Dictionary of individual reward components
108         reward_dict = {
109             "speed_reward": speed_reward,
110             "consistency_reward": consistency_reward,
111         }
112
113         return reward, reward_dict
```

Example 2 (b): *URDP* reward functions on Humanoid. Iteration 1 (before UABO), HNS: 1.428, Iteration 1 (after UABO), HNS: 2.132; Iteration 2 (before UABO), HNS: 2.481, Iteration 2 (after UABO), HNS: 2.646.

```
1    Iteration 1(before UABO):
2    def compute_reward(root_states: torch.Tensor, dt: float) -> Tuple[torch.Tensor, Dict[str, torch.Tensor]]:
3           # Extract the velocity of the humanoid's torso from the root states
4           velocity = root_states[:, 7:10]
5
6           # Compute the forward speed (assuming x-axis is the forward direction)
7           forward_speed = velocity[:, 0]
8
9           # Scale up the speed reward to provide stronger feedback
10          speed_reward_temp = 10  # Temperature parameter for speed reward
11          speed_reward = torch.exp(speed_reward_temp * forward_speed * dt)
12
13          # Optional: Add a reward for sustained running
14          sustained_speed_temp = 0.1  # Temperature parameter for sustained speed reward
15          sustained_speed_reward = torch.exp(sustained_speed_temp * forward_speed * dt)
16
17          # Combine the rewards
18          total_reward = speed_reward + sustained_speed_reward
19
20          # Return the total reward and the individual reward components
21          reward_dict = {
22              "speed_reward": speed_reward,
23              "sustained_speed_reward": sustained_speed_reward
24          }
25
26          return total_reward, reward_dict
27
28   ################################################################################################
29
30   Iteration 1(after UABO):
31   def compute_reward(root_states: torch.Tensor, dt: float) -> Tuple[torch.Tensor, Dict[str, torch.Tensor]]:
32          # Extract the velocity of the humanoid's torso from the root states
33          velocity = root_states[:, 7:10]
```

```
34
35          # Compute the forward speed (assuming x-axis is the forward direction)
36          forward_speed = velocity[:, 0]
37
38          # Scale up the speed reward to provide stronger feedback
39          speed_reward_temp = 10.4424  # Temperature parameter for speed reward
40          speed_reward = torch.exp(speed_reward_temp * forward_speed * dt)
41
42          # Optional: Add a reward for sustained running
43          sustained_speed_temp = 0.3606  # Temperature parameter for sustained speed reward
44          sustained_speed_reward = torch.exp(sustained_speed_temp * forward_speed * dt)
45
46          # Combine the rewards
47          total_reward = speed_reward + sustained_speed_reward
48
49          # Return the total reward and the individual reward components
50          reward_dict = {
51              "speed_reward": speed_reward,
52              "sustained_speed_reward": sustained_speed_reward
53          }
54
55          return total_reward, reward_dict
56
57  ####################################################################################################
58
59  Iteration 2(before UABO):
60  def compute_reward(root_states: torch.Tensor, dt: float) -> Tuple[torch.Tensor, Dict[str, torch.Tensor]]:
61          # Extract the velocity of the humanoid's torso from the root states
62          velocity = root_states[:, 7:10]
63          torso_height = root_states[:, 2]  # Torso height (z-axis)
64
65          # Compute the forward speed (assuming x-axis is the forward direction)
66          forward_speed = velocity[:, 0]
67
68          # Apply exponential transformation to the speed reward with a temperature parameter
69          speed_temp = 1  # Temperature parameter for speed reward
70          speed_reward = torch.exp(speed_temp * forward_speed) * dt
71
72          # Penalty for falling (torso height below a threshold)
73          fall_threshold = 0.8  # Minimum torso height to avoid falling
74          fall_penalty = -10.0 * (torso_height < fall_threshold).float() * dt
75
76          # Combine the rewards
77          total_reward = speed_reward + fall_penalty
78
79          # Return the total reward and the individual reward components
80          reward_dict = {
81              "speed_reward": speed_reward
82          }
83
84          return total_reward, reward_dict
85
86  ####################################################################################################
87
88  Iteration 2(after UABO):
89  def compute_reward(root_states: torch.Tensor, dt: float) -> Tuple[torch.Tensor, Dict[str, torch.Tensor]]:
90          # Extract the velocity of the humanoid's torso from the root states
91          velocity = root_states[:, 7:10]
92          torso_height = root_states[:, 2]  # Torso height (z-axis)
93
94          # Compute the forward speed (assuming x-axis is the forward direction)
95          forward_speed = velocity[:, 0]
96
97          # Apply exponential transformation to the speed reward with a temperature parameter
98          speed_temp = 0.7111  # Temperature parameter for speed reward
99          speed_reward = torch.exp(speed_temp * forward_speed) * dt
100
101          # Penalty for falling (torso height below a threshold)
102          fall_threshold = 0.8  # Minimum torso height to avoid falling
103          fall_penalty = -10.0 * (torso_height < fall_threshold).float() * dt
104
105          # Combine the rewards
106          total_reward = speed_reward + fall_penalty
107
108          # Return the total reward and the individual reward components
109          reward_dict = {
110              "speed_reward": speed_reward
111          }
112
113          return total_reward, reward_dict
```

## D.2 Case Study 2: the Redundancy in Reward Function Samples

This case study analyzes two representative examples from Eureka within a single iteration, visually demonstrating a major source of computational redundancy. Our analysis reveals that Eureka generates multiple semantically equivalent but syntactically varied reward functions within a single iteration, all sharing identical reward intensities.

For instance, in Example 3, although the two reward functions (Sample #9 and #15) from iteration 1 exhibit different textual expressions ("*forward_velocity_reward*" vs. "*velocity_reward*"), their underlying reward objectives and logic are fundamentally identical. This observation suggests that the apparent diversity among Eureka-generated samples may be primarily lexical rather than semantic. Effective filtering of such pseudo-diversity is therefore essential to eliminate redundant and computationally inefficient evaluations. Example 4 is a similar example.

Example 3: The reward function codes of the task Ant after Iteration 1. The scores of Sample #9 and Sample #15 are both 2.012.

```
1   Iteration 1 (sample #9):
2   def compute_reward(root_states: torch.Tensor, actions: torch.Tensor) -> Tuple[torch.Tensor, Dict[str, torch.Tensor]]:
3
4       # Define weight parameters
5       forward_velocity_temp: float = 1.0
6       action_penalty_temp: float = 0.01
7
8       # Extract forward velocity (x-axis velocity in the world frame)
9       forward_velocity = root_states[:, 7]  # Velocity along the x-axis
10
11      # Reward for forward velocity
12      forward_velocity_reward = forward_velocity * forward_velocity_temp
13
14      # Penalty for large actions to encourage energy efficiency
15      action_penalty = -torch.sum(torch.square(actions), dim=-1) * action_penalty_temp
16
17      # Total reward
18      reward = forward_velocity_reward + action_penalty
19
20      # Individual reward components
21      reward_dict = {
22          "forward_velocity_reward": forward_velocity_reward,
23          "action_penalty": action_penalty,
24      }
25
26      return reward, reward_dict
27
28  ################################################################################################
29
30  Iteration 1 (sample #15):
31  def compute_reward(root_states: torch.Tensor, actions: torch.Tensor) -> Tuple[torch.Tensor, Dict[str, torch.Tensor]]:
32
33      # Define weight parameters
34      velocity_temp: float = 1.0
35      action_penalty_temp: float = 0.01
36
37      # Extract the forward velocity (x-axis velocity)
38      forward_velocity = root_states[:, 7]  # x-axis velocity is at index 7
39
40      # Reward for moving forward fast
41      velocity_reward = forward_velocity * velocity_temp
42
43      # Penalty for large actions to encourage energy efficiency
44      action_penalty = -torch.sum(torch.square(actions), dim=-1) * action_penalty_temp
45
46      # Total reward
47      total_reward = velocity_reward + action_penalty
48
49      # Individual reward components for debugging and analysis
50      reward_components = {
51          "velocity_reward": velocity_reward,
52          "action_penalty": action_penalty,
53      }
54
55      return total_reward, reward_components
```

Example 4: The reward function codes of the task Ant after Iteration 1. The scores of Sample #3 and Sample #14 are both 0.059.

```
1   Iteration 1 (sample #3):
2   def compute_reward(object_rot: torch.Tensor, goal_rot: torch.Tensor, object_angvel: torch.Tensor) -> Tuple[torch.Tensor,
    Dict[str, torch.Tensor]]:
3       # Temperature parameters for reward components
4       orientation_temp: float = 1.0
5       angular_vel_temp: float = 0.1
6
7       # Compute the difference in orientation using quaternion distance
8       quat_diff = quat_mul(object_rot, quat_conjugate(goal_rot))
9       orientation_error = torch.norm(quat_diff[:, 1:4], dim=1)  # Ignore the scalar part for distance
10      orientation_reward = torch.exp(-orientation_temp * orientation_error)
11
12      # Penalize excessive angular velocity
13      angular_vel_penalty = torch.exp(-angular_vel_temp * angular_vel_magnitude)
14
15      # Combine rewards
16      total_reward = orientation_reward * angular_vel_penalty
17
18      # Return the total reward and individual components for debugging
19      reward_components = {
20          "orientation_reward": orientation_reward,
21          "angular_vel_penalty": angular_vel_penalty
22      }
23      return total_reward, reward_components
24
25  #########################################################################################################
26
27  Iteration 1 (sample #14):
28  def compute_reward(object_rot: torch.Tensor, goal_rot: torch.Tensor, object_angvel: torch.Tensor) -> Tuple[torch.Tensor,
    Dict[str, torch.Tensor]]:
29      # Temperature parameters for reward components
30      orientation_temp = 1.0
31      angular_velocity_temp = 0.1
32
33      # Compute the difference in orientation using quaternion distance
34      quat_diff = quat_mul(object_rot, quat_conjugate(goal_rot))
35      orientation_error = torch.norm(quat_diff[:, 1:4], dim=1)  # Ignore the scalar part for distance
36      orientation_reward = torch.exp(-orientation_temp * orientation_error)
37
38      # Penalize excessive angular velocity
39      angular_velocity_magnitude = torch.norm(object_angvel, dim=1)
40      angular_velocity_penalty = torch.exp(-angular_velocity_temp * angular_velocity_magnitude)
41
42      # Combine rewards
43      total_reward = orientation_reward * angular_velocity_penalty
44
45      # Return the total reward and individual components for debugging
46      reward_components = {
47          "orientation_reward": orientation_reward,
48          "angular_velocity_penalty": angular_velocity_penalty
49      }
50      return total_reward, reward_components
```

# E   Detailed Results

## E.1   Evaluation on Efficiency

Table 6 presents the comprehensive evaluation results across all tasks in the three benchmarks. The comparative analysis demonstrates that while achieving comparable SR or HNS to Eureka, *URDP* requires fewer simulation training episodes and LLM invocations in 92% of the experimental tasks, indicating superior sample efficiency and computational economy.

## E.2   Evaluation on the Performance of the Reward Function

Table 7 presents a comprehensive performance comparison of different methods across all tasks in three benchmarks. The results demonstrate that *URDP* consistently outperforms the baseline approaches while maintaining comparable or reduced requirements for both simulation training episodes and LLM invocations.

Table 6: *URDP* vs. SOTA with efficiency. *URDP* performed best in 92% of tasks in terms of NOE and NLC (bolded parts in the table).

| Benchmark | Environment | Text2Reward | | | Eureka | | | *URDP* | | |
|---|---|---|---|---|---|---|---|---|---|---|
| | | HNS | NOE↓ | NLC↓ | HNS | NOE↓ | NLC↓ | HNS | NOE↓ | NLC↓ |
| Isaac | Ant | 1.543 | 112 | 7 | 1.527 | 112 | 7 | 1.556 | **48** | **3** |
| | Cartpole | 1 | 16 | 1 | 1 | 16 | 1 | 1 | **15** | **1** |
| | BallBalance | 1 | 16 | 1 | 1 | 16 | 1 | 1 | **16** | **1** |
| | Quadcopter | 1.678 | 82 | 6 | 1.667 | 70 | 5 | 1.818 | **41** | **2** |
| | FrankaCabinet | 16.95 | 95 | 7 | 17 | 97 | 7 | 17.130 | **57** | **4** |
| | Humanoid | 2.305 | 16 | 1 | 2.306 | 16 | 1 | 2.646 | 52 | 3 |
| | Anymal | 1.095 | 87 | 6 | 1.113 | 91 | 6 | 1.2 | **47** | **2** |
| | AllegroHand | 2.176 | 121 | 8 | 2.162 | 95 | 6 | 2.182 | **40** | **4** |
| | ShadowHand | 1.805 | 111 | 8 | 1.786 | 105 | 7 | 1.817 | **33** | **2** |
| | | SR | NOE↓ | NLC↓ | SR | NOE↓ | NLC↓ | SR | NOE↓ | NLC↓ |
| Dexterity | BlockStack | 0.67 | 112 | 7 | 0.67 | 112 | 7 | 0.68 | **53** | **1** |
| | Kettle | 0.89 | 85 | 6 | 0.89 | 72 | 5 | 0.89 | **78** | **5** |
| | DoorCloseOutward | 0.96 | 83 | 6 | 0.9 | 72 | 5 | 0.97 | **37** | **2** |
| | DoorCloseInward | 1 | 71 | 5 | 1.0 | 78 | 5 | 1.0 | **34** | **2** |
| | SwingCup | 0.84 | 93 | 7 | 0.84 | 87 | 6 | 0.84 | **51** | **3** |
| | Switch | 0 | 58 | 5 | 0.0 | 58 | 5 | 0.02 | 76 | 5 |
| | TwoCatchUnderarm | 0 | 62 | 5 | 0.0 | 62 | 5 | 0.0 | **62** | **4** |
| | CatchUnderarm | 0.72 | 95 | 7 | 0.73 | 89 | 6 | 0.73 | **67** | **4** |
| | CatchAbreast | 0.66 | 88 | 6 | 0.66 | 83 | 6 | 0.67 | **54** | **4** |
| | DoorOpenInward | 0.04 | 73 | 5 | 0.04 | 69 | 5 | 0.06 | **67** | **4** |
| | PushBlock | 0.14 | 97 | 7 | 0.14 | 92 | 6 | 0.15 | **49** | **3** |
| | BottleCap | 0.88 | 110 | 8 | 0.88 | 96 | 7 | 0.89 | **67** | **4** |
| | ReOrientation | 0.32 | 75 | 6 | 0.31 | 66 | 5 | 0.33 | **58** | **4** |
| | CatchOver2Underarm | 0.91 | 77 | 6 | 0.9 | 74 | 5 | 0.93 | **57** | **3** |
| | LiftUnderarm | 0.89 | 88 | 6 | 0.89 | 86 | 6 | 0.89 | **78** | **4** |
| | Over | 0.92 | 82 | 6 | 0.92 | 71 | 5 | 0.92 | **41** | **3** |
| | Pen | 0.85 | 95 | 7 | 0.85 | 97 | 7 | 0.85 | **78** | **4** |
| | DoorOpenOutward | 1 | 76 | 5 | 1.0 | 75 | 5 | 1.0 | **46** | **3** |
| | Scissors | 1 | 76 | 5 | 1.0 | 77 | 5 | 1 | **44** | **3** |
| | GraspAndPlace | 0.75 | 93 | 6 | 0.75 | 85 | 6 | 0.77 | **59** | **4** |
| | | SR | NOE↓ | NLC↓ | SR | NOE↓ | NLC↓ | SR | NOE↓ | NLC↓ |
| ManiSkill2 | LiftCube | 0.906 | 112 | 7 | 0.905 | 96 | 6 | 0.906 | **15** | **1** |
| | PickCube | 0.879 | 128 | 8 | 0.884 | 112 | 7 | 0.885 | **20** | **1** |
| | TurnFaucet | 0.799 | 96 | 6 | 0.800 | 96 | 6 | 0.801 | **34** | **2** |
| | OpenCabinetDoor | 0.865 | 96 | 6 | 0.861 | 96 | 6 | 0.866 | **31** | **2** |
| | OpenCabinetDrawer | 0.633 | 112 | 7 | 0.632 | 96 | 6 | 0.638 | **43** | **2** |
| | PushChair | 0.657 | 96 | 6 | 0.654 | 96 | 6 | 0.657 | **51** | **2** |

Notably, `URDP` achieves superior performance to human-designed reward functions in 89% of the experimental tasks, highlighting the significant potential of automated reward design methodologies.

Table 7: Task-wise comparison of `URDP` with other methods. `URDP` outperforms the compared methods on 89% of the tasks.

| Benchmark | Environment | Sparse | Human | Text2Reward | | | Eureka | | | *URDP* | | |
|---|---|---|---|---|---|---|---|---|---|---|---|---|
| | | HNS↑ | HNS↑ | HNS↑ | NOE | NLC | HNS↑ | NOE | NLC | HNS↑ | NOE | NLC |
| Isaac | Ant | 0 | 1 | 0.772 | 48 | 3 | 0.828 | 48 | 3 | **1.556** | 48 | 3 |
| | Cartpole | 0 | 1 | 1 | 15 | 1 | 1 | 15 | 1 | **1** | 15 | 1 |
| | BallBalance | 0 | 1 | 1 | 16 | 1 | 1 | 16 | 1 | **1** | 16 | 1 |
| | Quadcopter | 0 | 1 | 1.041 | 41 | 3 | 1.25 | 41 | 3 | **1.818** | 41 | 2 |
| | FrankaCabinet | 0 | 1 | 5.4 | 57 | 4 | 4.8 | 57 | 4 | **17.130** | 57 | 4 |
| | Humanoid | 0 | 1 | 2.217 | 52 | 4 | 2.306 | 52 | 4 | **2.646** | 52 | 3 |
| | Anymal | 0 | 1 | 0.317 | 47 | 3 | 0.545 | 47 | 3 | **1.2** | 47 | 2 |
| | AllegroHand | 0 | 1 | 1.196 | 40 | 3 | 1.594 | 40 | 3 | **2.182** | 40 | 4 |
| | ShadowHand | 0 | 1 | 1.034 | 33 | 3 | 1.115 | 33 | 3 | **1.817** | 33 | 2 |
| | | SR↑ | SR↑ | SR↑ | NOE | NLC | SR↑ | NOE | NLC | SR↑ | NOE | NLC |
| Dexterity | BlockStack | 0 | 0.69 | 0.11 | 53 | 4 | 0.12 | 53 | 4 | 0.679 | 53 | 1 |
| | Kettle | 0 | 0.02 | 0.89 | 78 | 5 | 0.89 | 78 | 5 | **0.89** | 72 | 5 |
| | DoorCloseOutward | 0.15 | 0.06 | 0.57 | 37 | 3 | 0.64 | 37 | 3 | **0.968** | 37 | 2 |
| | DoorCloseInward | 0 | 1 | 0.74 | 34 | 3 | 0.83 | 34 | 3 | **1** | 34 | 2 |
| | SwingCup | 0 | 0 | 0.62 | 51 | 4 | 0.53 | 51 | 4 | **0.84** | 51 | 3 |
| | Switch | 0 | 0 | 0 | 76 | 5 | 0.01 | 76 | 5 | **0.02** | 76 | 5 |
| | TwoCatchUnderarm | 0 | 0 | 0 | 62 | 5 | 0 | 62 | 5 | **0** | 62 | 4 |
| | CatchUnderarm | 0 | 0.51 | 0.58 | 67 | 5 | 0.63 | 67 | 5 | **0.73** | 67 | 4 |
| | CatchAbreast | 0 | 0.37 | 0.27 | 54 | 5 | 0.34 | 54 | 5 | **0.67** | 54 | 4 |
| | DoorOpenInward | 0 | 0.03 | 0 | 67 | 5 | 0 | 67 | 5 | **0.06** | 67 | 4 |
| | PushBlock | 0 | 0.01 | 0.05 | 49 | 4 | 0.05 | 49 | 4 | **0.15** | 49 | 3 |
| | BottleCap | 0.91 | 0.91 | 0.21 | 67 | 5 | 0.25 | 67 | 5 | **0.89** | 67 | 4 |
| | ReOrientation | 0.01 | 0.02 | 0.25 | 58 | 4 | 0.28 | 58 | 4 | **0.33** | 58 | 4 |
| | CatchOver2Underarm | 0 | 0.87 | 0.81 | 57 | 4 | 0.81 | 57 | 4 | **0.93** | 57 | 3 |
| | LiftUnderarm | 0 | 0.37 | 0.83 | 78 | 6 | 0.85 | 78 | 6 | **0.89** | 78 | 4 |
| | Over | 0 | 0.9 | 0.54 | 41 | 4 | 0.61 | 41 | 4 | **0.92** | 41 | 3 |
| | Pen | 0.01 | 0.74 | 0.67 | 78 | 6 | 0.63 | 78 | 6 | **0.85** | 78 | 4 |
| | DoorOpenOutward | 0.02 | 0.85 | 0.76 | 46 | 3 | 0.87 | 46 | 3 | **1** | 46 | 3 |
| | Scissors | 0.99 | 0.96 | 0.73 | 44 | 3 | 0.69 | 44 | 3 | **1** | 44 | 3 |
| | GraspAndPlace | 0 | 0.87 | 0.41 | 59 | 4 | 0.43 | 59 | 4 | 0.77 | 59 | 4 |
| | | SR↑ | SR↑ | SR↑ | NOE | NLC | SR↑ | NOE | NLC | SR↑ | NOE | NLC |
| ManiSkill2 | LiftCube | 0.143 | 0.543 | 0.531 | 15 | 1 | 0.356 | 15 | 1 | **0.906** | 15 | 1 |
| | PickCube | 0.131 | 0.479 | 0.497 | 20 | 2 | 0.434 | 20 | 2 | **0.885** | 20 | 1 |
| | TurnFaucet | 0 | 0.598 | 0.631 | 34 | 3 | 0.516 | 34 | 3 | **0.801** | 34 | 2 |
| | OpenCabinetDoor | 0.028 | 0.651 | 0.713 | 31 | 2 | 0.575 | 31 | 2 | **0.866** | 31 | 2 |
| | OpenCabinetDrawer | 0 | 0.37 | 0.519 | 43 | 3 | 0.478 | 43 | 3 | **0.64** | 43 | 2 |
| | PushChair | 0 | 0.334 | 0.432 | 51 | 4 | 0.336 | 51 | 4 | **0.657** | 51 | 2 |

# F Proofs

## F.1 Determination of the Kernel Function

**Theorem 1.** *The kernel function equation 6 satisfied the properties of symmetry and positive semi-definiteness.*

*Proof.* The property of symmetry is obvious. Now we prove the positive semi-definiteness based on the properties of the Matern kernel equation 4. For any given finite set of sample points $\tilde{p}^{(1)}, \tilde{p}^{(2)}, \cdots, \tilde{p}^{(n)}$, we denote the corresponding kernel matrix as

$$\tilde{K}_{ij} = \tilde{k}(\tilde{p}^{(i)}, \tilde{p}^{(j)}). \tag{10}$$

By performing a coordinate scaling transformation on the sample points, we obtain new sample points

$$p^{(i)} = (\tilde{p}_1^{(i)}/l_1, \cdots, \tilde{p}_d^{(i)}/l_d), \ i = 1, 2, \cdots, n. \tag{11}$$

And the Matern kernel matrix is

$$K_{ij} = k(p^{(i)}, p^{(j)}) \tag{12}$$

Since Matern kernel equation 4 is positive semi-definite, the kernel matrix equation 12 constructed from the transformed points with Matern kernel is positive semi-definite. Furthermore, the kernel matrix of equation 6 is essentially equivalent to the Martern kernel matrix equation 12 computed on the transformed sample points, i.e.

$$r_{\text{new}}(\tilde{p}^{(i)}, \tilde{p}^{(j)}) = r(p^{(i)}, p^{(j)}) \tag{13}$$

$$\tilde{K}_{ij} = \tilde{k}(\tilde{p}^{(i)}, \tilde{p}^{(j)}) = f_\nu(r_{\text{new}}) = f_\nu(r) = K_{ij}. \tag{14}$$

Therefore, $\tilde{K}$ is positive semi-definite. $\qquad\square$

In the newly defined weighted distance equation 5, a larger $l_i$ in directions with more rapid variations can increase the possibility of exploration, while a smaller $l_i$ in directions with smoother variations will reduce exploration and emphasize the exploitation of information from previously sampled points.

### F.2 Convergence analysis of the Uncertainty-accelerated Expected Improvement (uEI)

The basic definitions and theorems have been defined to analyze the convergence rate of Bayesian optimization (Bull, 2011). Here, we briefly restate some of the key definitions required. Let $\mathcal{X} \subset \mathbb{R}^d$ be a compact set with non-empty interior. For a function $f : \mathcal{X} \to \mathbb{R}$ to be minimized, $K_\theta$ is the correlation kernel for the function $f$ prior distribution $\pi$ with length-scales $\theta$. $\mathcal{H}_\theta(\mathcal{X})$ is the reproducing-kernel Hilbert space of $K_\theta$ on $\mathcal{X}$. Let $\mathbb{P}_f^u$ and $\mathbb{E}_f^u$ denote the probability and expectation operators when minimizing the fixed function $f$ using strategy $u$. The loss suffered over the ball $B_R$ in $\mathcal{H}_\theta(\mathcal{X})$ after $n$ steps by a strategy $u$ is defined as,

$$L_n(u, \mathcal{H}_\theta(\mathcal{X}), R) := \sup_{\substack{f \in \mathcal{H}_\theta(\mathcal{X}) \\ \|f\|_{\mathcal{H}_\theta(\mathcal{X})} \leq R}} \mathbb{E}_f^u \left[ f(x_n^*) - \min f \right] \tag{15}$$

where $x_n^*$ is the estimated minimum of $f$.

It is proved that the strategy expected improvement converges at least at a rate $n^{-(\nu \wedge 1)/d}$, up to logarithmic factors, where $\nu$ is the parameter in the Matern kernel (Bull, 2011).

**Theorem 2.** *Assume that the function $f$ depends only on $m$ input variables, $m < d$, and remains constant along the other $d-m$ directions. Under such an assumption, with an appropriate choice of weighted parameters, the Uncertainty-accelerated Expected Improvement converges at least at the rate $n^{-(\nu \wedge 1)/m}$, up to logarithmic factors, where $\nu$ is the parameter in the Matern kernel.*

*Proof.* From the proof of expected improvement convergence rate (Bull, 2011), we observe that the parameter $d$ in convergence rate estimation is actually the dimensionality of the sampling space. And the conclusion holds based on the condition that $\{x_n\}$ is a quasi-uniform sequence in a region of interest (Narcowich et al., 2003). Without loss of generality, let us assume that the function $f$ depends on dimensions $i_1$ to $i_m$, and is invariant with respect to dimensions $i_{m+1}$ to $i_d$. For the uEI strategy, let

$$\lambda_j = \begin{cases} 0, & j = 1, \cdots, m \\ \infty, & j = m+1, \cdots, d \end{cases} \tag{16}$$

Consequently, any exploration in the directions of dimensions $i_{m+1}$ to $i_d$ will be discouraged. The effective dimensionality of the sampling space decreases from $d$ to $m$, which leads to an improved convergence rate of at least $n^{-(\nu \wedge 1)/m}$. $\qquad\square$

Although our theorem has focused on the limiting case in which $f$ is entirely independent of certain directions, it illustrates how applying weighted constraints allows for dimension-specific treatment within the sampling space, thus enhancing the efficiency of the algorithm.

# G   Additional Analysis

## G.1   Uncertainty and Reward Shaping

To validate the role of high-uncertainty reward components, we conducted ablation studies by removing these components from the reward function. Figure 12 presents comparative cases between the original reward functions ($R$) and its ablated counterparts ($R$ w.o. $r_{u\uparrow}$). Our analysis reveals two key findings: (1) the removal of high-uncertainty components leads to significant performance degradation, with respective decreases of 19%, 83%, and 51% in HNS/SR metrics; and (2) reward functions retaining these components demonstrate accelerated discovery of critical states during early RL training phases, effectively reducing inefficient exploration. These results collectively demonstrate the crucial function of high-uncertainty components in both final performance and training efficiency. These results suggest that high-uncertainty reward components contribute positively to reward shaping and play an essential role in guiding effective policy learning.

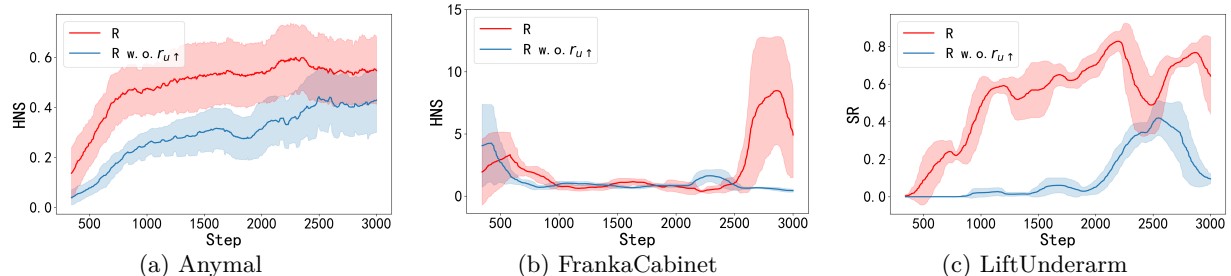

Figure 12: The comparison between $R$ and $R$ w.o.$r_{u\uparrow}$ suggests that the high-uncertainty reward components $r_{u\uparrow}$ (*dof_vel_penalty* in Anymal, *progress_reward* in FrankaCabinet, *drop_penalty* in LiftUnderarm) contribute to reward shaping during the policy learning.

## G.2   LLM Alternatives

**URDP with Qwen2.5 and Llama3**. In Figure 13, we compare the performance of **URDP** with DeepSeek-v3-241226 (the results reported in the paper), **URDP** with Qwen2.5 (qwen-max-0919) (Qwen et al., 2025), and Llama3 (llama-v3-70b-instruct) (Dubey et al., 2024). These results demonstrate the consistency of the effect of **URDP** on different LLMs and eliminate concerns that the differences in the capabilities of LLMs themselves may affect the results.

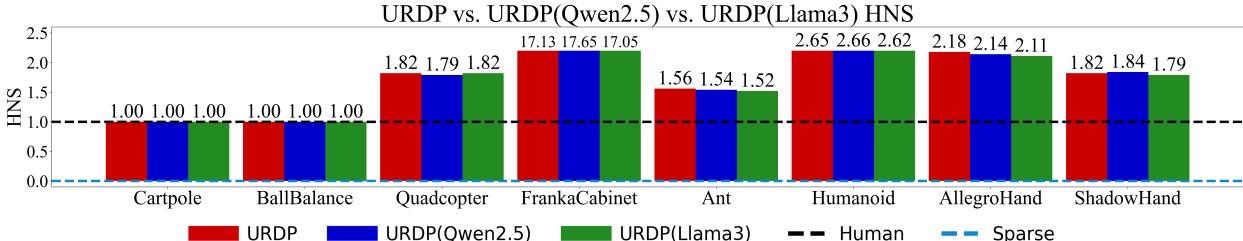

Figure 13: **URDP** demonstrates consistent performance across different LLMs.

# H   Limitation and Discussion

In this work, we investigate efficient automated reward design methodologies based on large language models (LLMs). However, constrained by the inherent limitations of LLMs in spatial reasoning capabilities, our approach, like other comparable methods, faces challenges in addressing scenario-specific constraints during reward formulation. A representative case emerges in "grasping" tasks where environmental obstacles may restrict robotic manipulation paths, constraints that should ideally be reflected in reward design. While

providing detailed environmental descriptions in prompts may partially mitigate this issue, a more fundamental solution would involve integrating video-language models (VLMs) into the reward design framework. VLMs demonstrate superior spatial perception capabilities that could enrich the understanding of RL task objectives, environmental constraints, and reward composition. Nevertheless, incorporating VLMs introduces new challenges regarding computational scalability during reward design and tuning processes. We therefore identify this as a critical yet underexplored research direction worthy of systematic investigation.

To maintain simplicity in presenting our work, we employ the base capabilities of large language models without sophisticated inference-time enhancement techniques (e.g., chain-of-thought, test-time training). However, advanced reasoning techniques have demonstrated significant improvements in handling complex logical tasks, as evidenced in code generation and mathematical reasoning domains. We posit these methods would similarly enhance reward function code design. Ultimately, substantial exploration potential remains for large language model techniques in automated reward design.

