# OpenReview forum: "Uncertainty-aware Reward Design Process"
_TMLR — Accepted by TMLR_

### Review · Reviewer_vsaA · 2025-07-16

**Summary Of Contributions:**

This paper proposes URDP, a novel framework for automating reward function design in reinforcement learning (RL) using large language models (LLMs) and uncertainty-aware Bayesian optimization. The authors argue that conventional reward engineering and inverse RL approaches are either too reliant on human expertise or costly in terms of demonstration data, and that recent LLM-based approaches still suffer from inefficiencies in reward quality and optimization. The proposed URDP decouples reward logic construction and numerical hyperparameter optimization, introducing a bi-level architecture supported by uncertainty quantification and hybrid optimization. The method is evaluated across 35 tasks in three benchmark environments.

**Audience:**

Yes

**Audience Explanation:**

Given the novelty of combining LLM reasoning with reward function generation and optimization, and the growing interest in automating RL pipelines, this paper would appeal to subsets of the TMLR community, particularly those working on: Reinforcement learning automation, LLM-based control or code generation, Program synthesis and reward modeling, Optimization under uncertainty.

**Claims And Evidence:**

Yes

**Claims Explanation:**

The authors provide comprehensive experimental results.

**Requested Changes:**

1. The paper fails to clearly define the optimization problem being solved. It is unclear what the exact objective functions are and how the components of reward generation and tuning are jointly or separately optimized. A formal problem definition would be helpful to ground the framework in established RL terminology.

2. The manuscript mentions prior work involving evolutionary search and introduces uncertainty-aware Bayesian optimization but does not explicitly compare or connect these paradigms. It is crucial to explain how URDP differs from, improves upon, or integrates with these approaches, especially in how LLMs are used within this hybrid framework.

3. Currently, it is unclear what constitutes a "high-quality" reward function. Is it measured by policy performance, training sample efficiency, convergence speed, or interpretability? Without this clarity, the claims in the abstract and results lack a solid basis.

4. The explanation of both RL fundamentals and the specifics of the reward design problem is insufficient in the preliminaries. For a reader unfamiliar with recent LLM-RL research, the paper may be difficult to follow.

5. Figure 1 is visually appealing but does not make clear how URDP fits into the overall RL training loop.

6. Several parts of the paper suffer from vague phrasing and a lack of precision.

7. The abstract does not effectively communicate the problem, proposed method, or key results.

---

> ### Author Response · Authors · 2025-08-19
>
> We sincerely thank you for your detailed and constructive review.
>
> Q1: A formal problem definition.
>
> A: We appreciate the reviewers’ suggestions. In the uploaded revised paper, we formally defined RL and RDP in Sect. 3.
>
> Q2: Evolutionary search and How LLMs are used.
>
> A: A significant distinction between URDP and Eureka, a conventional LLM-based evolutionary search technique, lies in the decoupling of the optimization process. Eureka employs fully LLM-based evolutionary search, with both reward component design and reward strength optimization handled by the LLM. In contrast, URDP retains LLM-driven evolutionary search only in the outer loop for reward component optimization. Moreover, it replaces the inner loop’s reward intensity optimization with uncertainty-aware Bayesian optimization (UABO) to circumvent the inherent limitations of LLMs in numerical optimization (such as insufficient parameter tuning precision). We clarified this difference in the updated Section 2. We pointed out the use of LLM throughout the framework in the updated Section 4.1, and we also noted that all prompt words are provided in Appendix A.
>
> Q3: “High-quality” rewards.
>
> A: “High-quality” rewards refer to the policy performance, quantified by specific metrics: Success Rate (SR) for Dexterity and ManiSkill2 tasks, and Human Normalized Score (HNS) for Isaac tasks. These metrics directly reflect the performance of policies trained with the reward function in tasks. To avoid ambiguity, we have revised potentially confusing expressions in the original manuscript and used "high-performance rewards" to refer to them.
>
> Q4: The explanation of RL fundamentals and the reward design problem.
>
> A: We formally described reinforcement learning in the updated Section 3. Following the previous work (Eureka), we formally defined the reward design problem. We added the content to help readers understand the preblem.
>
> Q5: How URDP fits into the overall RL training loop in Figure 1?
>
> A: We have updated Figure 1 in the revised manuscript. In the inner loop, candidate reward functions are passed to the RL training module (using standard PPO and SAC algorithms) to evaluate the performance of the generated reward functions through RL training.
>
> Q6: Vague phrasing.
>
> A: We have thoroughly revised this article to ensure accuracy and have added necessary information to fill in any gaps.
>
> Q7: The abstract.
>
> A: We have revised the abstract in the updated version of the paper to ensure that the problem to be solved is clearly presented, and that the methods and results of the paper are summarized.

---

### Review · Reviewer_iGv2 · 2025-07-30

**Summary Of Contributions:**

**Summary:**

This work proposes URDP, a novel uncertainty-aware reward design paradigm for RL that introduces a bi-level optimization architecture including reward component generation and hyperparameter optimization. The method leverages LLMs' generative capabilities to produce diverse reward components in the outer loop, while employing Uncertainty-Aware Bayesian Optimization (UABO) to learn optimal hyperparameters in the inner loop, with a novel uncertainty quantification mechanism strategically selecting the most promising components to reduce computational costs. Through comprehensive experiments across three benchmarks, the authors demonstrate URDP's effectiveness, particularly highlighting its hybrid approach that combines LLMs' creative reward formulation with Bayesian optimization's numerical precision.


**Strengths:**

1. This work thoughtfully addresses the critical challenge of reward function design in reinforcement learning.
2. The choice of Bayesian optimization proves particularly valuable for handling numerical optimization tasks where LLMs typically struggle.
3. Through comprehensive benchmarking experiments and carefully designed ablation studies, the manuscript demonstrates both the method's effectiveness and the necessity of its individual components.

**Weaknesses:**

1. Some methodological descriptions could benefit from further clarification:
    - the difference between $R_i$ and $r_{i,1}, \dots, r_{i,m}$.
    - the descriptions of the $(x,y)$ data used in Bayesian Optimization.
2. The approach also involves several hyperparameters (e.g., smoothness $\nu$ and length scale $\ell$), and discussing UABO's sensitivity to these parameters would strengthen the technical discussion.


**Question:**
1. Would you clarify which specific LLM performed the textual/semantic similarity calculations for uncertainty quantification?
2. Could you elaborate on the technique used to balance exploration-exploitation in Section 4.2, and how you determined the iteration counts for the inner versus outer optimization loops?
3. To what extent does the choice of generative LLM (e.g., Qwen, LLaMA, or GPT-4 Omni) influence final performance?
4. Might you explain the meaning of "NOE" and the specific simulation context where this was applied?

**Audience:**

Yes

**Audience Explanation:**

This paper's novel integration of LLMs with Bayesian optimization for automated reward design addresses a fundamental RL challenge, directly appealing to researchers in reinforcement learning, Bayesian methods, and generative modeling through its demonstrable performance gains across established benchmarks.

**Broader Impact Concerns:**

No ethical concerns are noted.

**Claims And Evidence:**

Yes

**Claims Explanation:**

The claims are convincingly supported through comprehensive benchmarking across three domains, rigorous ablation studies validating each component's necessity, explicit efficiency analysis showing computational gains, and systematic comparisons against established baselines demonstrating consistent performance advantages.

**Requested Changes:**

I have one suggestion regarding the algorithm's positioning: It would enhance readability to relocate Algorithm 1 to a later section. Currently, it appears before readers encounter essential definitions like the distinction between $R_i$ and $r_{i,1}, \dots, r_{i,m}$, the composition of dataset $D$, and the specific prompting methodology. Presenting these foundations first would better prepare readers to understand the algorithm's mechanics and significance.

---

> ### Author Response · Authors · 2025-08-19
>
> We sincerely thank you for your detailed and insightful review.
>
> Q1: $R_i$ and $r_{i,m}$
>
> A: We have globally unified the notation throughout the revised manuscript: $R$ denotes a reward function, and $r$ denotes a reward component. Specifically, $R_i$ represents the reward function in the $i$-th sample, and $r_{i,m}$ is the $m$-th component within $R_i$. This clarification has been added in Section 3 and in Section 4.2.
>
> Q2: $(x, y)$ in BO
>
> A: We added the description in Section 4.3 of the updated paper. In our context, $y$ is the policy success rate from the simulation RL training and $x$ is the reward intensity.
>
>
> Q3: UABO’s sensitivity to the hyperparameters.
>
> A: We supplemented a series of experimental results on hyperparameters in Appendix C.2 of the uploaded revised paper to strengthen the technical discussion.
>
> Q4: Specific LLM in uncertainty quantification.
>
> A: As specified in Appendix C.1 “Implementation Details”, we use the BGE-M3 model for semantic similarity, as it provides strong multilingual, multifunctional embeddings well-suited for structured text like reward components. We use difflib.SequenceMatcher from Python’s standard library to compute surface-level text similarity.
>
> Q5: Exploration-exploitation balance and iteration counts
>
> A: In Algorithm 1, the maximum number of inner loops can be adaptively adjusted according to $N_{inner} \cdot U_{R_i}$. For reward components with high uncertainty, they will receive more iterations in order to explore better hyperparameter configurations. We supplemented this description in the updated Section 4.2. For iteration counts: (1) We set $N_{outer}=10$ based on experiments showing convergence within this range (updated in Appendix C.2, Fig. 8a). (2) We set $N_{inner}=10$ based on empirical needs across tasks. The hyperparameters are discussed in Appendix C.2.
>
> Q6: The choice of LLMs.
>
> A: In Appendix G.2, we report the experimental results that URDP using the default DeepSeek-v3-241226 and Qwen-max-0919. We added a result using LLaMA-v3-70B-Instruct in the updated manuscript since they have similar parameter scales. Results show URDP performs consistently across all three LLMs, indicating its robustness to different generators. A pointer to Appendix G.2 has been added in Section 5 to help readers locate the comparison.
>
> Q7: The meaning of NOE and the simulation context
>
> A: NOE (Number of Evaluations) refers to the total number of RL simulations performed across all reward candidates during optimization. The simulation contexts correspond to the environments used in our experiments: ShadowHand and Anymal in IsaacGym, Dexterity’s bimanual tasks, and Franka robot tasks in ManiSkill2.
>
> Q8: Relocate Algorithm 1
>
> A: Thank you for your suggestion. We have moved Algorithm 1 to the end of Section 4.3 for better readability in the revised manuscript.

---

### Review · Reviewer_gtm5 · 2025-08-05

**Summary Of Contributions:**

This paper propose a extension to the LLM generated reward design field by proposing a separation scheme between reward function generation and reward weight tuning. The authors use LLM to design specific reward functions and use Bayesian Optimization to find the best reward weights for each reward. Uncertainty qualification is applied for enhancing the reward function generation as well as improve the BO optimization step with a uncertainty aware optimization methods. The method demonstrates superior performance and efficiency against baselines across multiple benchmarks. Extensive case study and ablation study signifies the effectiveness of the proposed method, bundled with adequate convergence analysis.

## Strength
1. Interesting topic and solid approach. The approach is adequately justified and implemented. The separate of reward design and weight optimization makes sense.

2. Solid benchmark performance against baselines across multiple benchmarks.


## Weakness
1. The writing of this paper can be greatly improved. I find the explanation in the method section a bit unclear. For example, what is feedback prompt and what's the approach used here? I.e., how to judge from the simulation and give feedback to LLM for calling again?
2. The authors claim current methods has "(1) insufficient reasoning capability for reward logic derivation, and (2) inadequate exploration of novel reward components". But it is not very clear how the proposed method address these problems effectively. What experiments is demonstrating reward logic derivation and novel reward components? I am aware of Fig. 7, how does this directly imply novel reward components? For starter, what are these reward terms? I think big claims need more experiments and analysis to demonstrate.
3. The authors claim in the abstract that "evolutionary search paradigm demonstrates inefficient utilization of simulation resources,
resulting in prohibitively lengthy design cycles with disproportionate computational overhead" This gave me a falsehood of expectation that the authors have tricks to address this more directly. But I couldn't see it from the method section or am I missing something? Except from directly doing BO over the parameters, I don't see special treatment of the simulation. If so, the claim sounds a bit overreaching.
4. For the method itself, I find it a bit hard to follow. What is \\( y\\) in this context? What is the objective trying to achieve? Suppose you have a reward function generated by LLM, and you have an optimization trying to find the best reward parameter, then what is EI trying to achieve here? Please write the method section more rigorously.
5. Since this is related to LLM, what is the exact LLM tool you use for all the experiments including the baselines? As LLM keeps evolving, the ability to generate evolves too. Only through a fair comparison can we draw meaningful conclusions.


### Minor Issues
1. In fig 1, in middle box, is this supposed to say Uncertainty Quantification? The fig itself is of low quality.

2. \\( U_{R_i} == U(r_i)?\\)

**Audience:**

Yes

**Audience Explanation:**

Designing reward function and finetuning reward has been a long standing obstacle in the robotics learning community. This work automates the reward designing process, which looks quite interesting and promising.

**Broader Impact Concerns:**

No broader impact concern as far as I can tell.

**Claims And Evidence:**

Yes

**Claims Explanation:**

The experiments shown clearly demonstrate the effectiveness against baselines.

**Requested Changes:**

1. The citation format is very distracting. Please change to the correct format with parenthesis and only use NAME, (YEAR) when it is used as a noun.

2. The overall texts looks overwhelmingly generated by language models. I suggest the authors render the texts more carefully. Some sentences I think are overly decorated, such as
i.  (section 2) Our proposed framework decouples confounding factors in reward design to reduce cognitive load and incorporates uncertainty quantification to guide LLMs toward more focused analysis and refinement of reward logic relevance.
ii.  (4.1) insufficient attention being paid to the optimization of reward components in agent learning.

3. Although I recognize the baseline method (Eureka) have a similar comparison scheme, but it seems that both this work and Eureka claim that LLM can design better rewards than humans. However, given that the current robot community is yet to embrace such generation scheme, I find that claim a bit overreaching. For example, what if we compare LLM generated reward to train an imitation agent and compute joint tracking performance, versus reward designed by humans? Such comparison might be more meaningful here since the metric used, Human Normalized Score, does not make sense to me, or I need further explanation from the authors how this is a valid metric to compare to. Success rate is a good metric, so I still think this work presents a meaningful improvement over reward function genereation.

---

> ### Author Response · Authors · 2025-08-19
>
> We sincerely appreciate your valuable and detailed review.
>
> Q1: Feedback prompt and the approach.
>
> A: The refinements made by LLMs to the reward components are driven by the feedback prompt, which is provided in Appendix A (prompt 2). The feedback prompt covers: (i) simulation results of the current reward function, (ii) uncertainty of each component, and (iii) rule-based editing suggestions (retain, revise, remove) to the components. We have added the detailed description in the updated Section 4.1.
>
> Q2: Feedback mechanism from simulation to LLM.
>
> A: During the simulation, URDP extracts reward components’ extreme values, means, and overall policy performance (e.g., success rates), then compute uncertainty scores, standard deviations, and extreme deviations. Based on these, rule-based recommendations (defined in the feedback prompt) are generated to guide the next LLM iteration.
>
> Q3: Reward logic derivation
>
> A: The “reward logic” refers to the reward components in the reward function. To avoid any ambiguity in understanding, we modified this expression in Section 2 of the updated paper.  “Despite their success, these methods generally do not separate defining reward components and tuning its parameters, which may reduce the efficiency of automated reward design.”
>
> Q4: Novel reward components in Fig. 7.
>
> A: We discussed the potential correlation between uncertainties and novel reward components in Section 5.5 Disc-2 and Appendix G.1. In Fig.7, the y-axis gives the Pearson correlation between generated reward and human-designed reward, while correlation<0 means the reward component is a novel reward. We can observe that all novel reward are of higher uncertainties. Red “×” indicates components worse than human-designed; blue dots indicate better ones. Lower vertical position means lower correlation with human designs. Components in the lower-right region are novel and effective (e.g., shadowhand’s angvel\_penalty, distance\_penalty; anymal’s progress\_reward; frankacabinet’s drop\_penalty). Figure caption has been updated for clarity. We enumerated these reward components in the updated Fig. 7 and Appendix G.1.
>
> Q5: Evolutionary search paradigm.
>
> A: The “evolutionary search paradigm” actually refers to relying entirely on LLMs to optimize all aspects of the reward function (including reward components and reward intensities). To avoid any ambiguity in understanding, we use “LLM-only optimization” in the updated paper to refer to the method being compared. We verified in our experiments that LLMs perform poorly in numerical optimization (Section 5.4 Abl-3). Therefore, we employed LLM-based evolutionary search for reward component optimization, and improved Bayesian optimization for hyperparameter optimization.
>
>
> Q6: EI objective and $ y $.
>
> A: The updated Section 4.3 clarifies: $ y $ is the simulation-derived performance (e.g., success rate), and the objective is to maximize $ y = f(\theta) $, where $ \theta $ represents reward parameters. Expected Improvement (EI) selects the next $ \theta $ by maximizing the expected gain over the current best outcome.
>
> Q7: Exact LLM tool.
>
> A: Section 5.1 specifies that all experiments use DeepSeek-v3-241226. Appendix G.2 compares with other LLMs and shows minor differences, confirming the robustness of our method across models.
>
> Q8: Typo in Fig 1.
>
> A: Thank you for pointing out the issue with the image. We have reprocessed the images in the revised paper.
>
> Q9: Notations of $U_{R_i}$ and $U_{r_i}$.
>
> A: We have globally unified the notation throughout the revised manuscript. We revised $U(R_i)$ to $U_{R_i} $ and $U(r_i)$ is modified to $U_{r_i}$. Here, $R_i$ is a reward function. $r_i$ represents its reward components, and $i$ denotes the sample index. $U_{R_i}$ and $U_{r_i}$ represent uncertainty scores of the reward function and the corresponding components, as defined in Section 4.2.
>
> Q10: Citation format.
>
> A: Citation formatting has been revised throughout the revised paper.
>
> Q11: Writing.
>
> A: We revised the decorated sentences throughout the paper for clarity and conciseness.
>
> Q12: Can LLMs outperform human-designed rewards?
>
> A: We have added necessary limitations to avoid misinterpretation. Our results and Eureka’ work demonstrate that LLMs can generate competitive or better rewards in certain conventional tasks, reducing manual effort.
>
> Q13: Validity of HNS.
>
> A: Since each task in issac varies in semantic meaning and scale, HNS normalizes performance against human-designed rewards, enabling fair cross-task and cross-method comparisons (including with Eureka). For dexterity tasks, we use success rate (binary evaluation). Section 5.2 has been revised to clarify this metric.

---

### Decision · Action_Editor_QNDK · 2025-09-25

**Recommendation:** Accept with minor revision

**Additional Comments:**

While the authors have already updated their paper during the discussion phase, the paper still requires improvements in several aspects, mostly already mentioned by the reviewers. In particular:
* Improve the writing and clarity of the presentation and problem setting: E.g., it is not explained what "reward components" are. The current definition of the reward design process does not specify the feedback available to the process and also does not demand sample efficiency.
* Ensure consistency of notation, e.g., $RDP$ vs $rdp$; make sure all used symbols are carefully introduced, e.g., $U_{R_i}$ when it first appears. This is also an issue in Algorithm 1 for example.
* Explain concepts when they are first mentioned, e.g., "self-consistency" is only explained rather late.
* Be more precise, e.g., "how to filter out potentially problematic reward function samples..." is unclear in the sense that your use "problematic" likely to mean "redundant" or "uninformative". Similarly, in Section 5.3, you say for instance "... achieves optimal performance..." but looking at the referred figure none of the curves looks like it is converged (so what is "optimal" supposed to mean?)
* Quoting directly from a review: "The overall texts looks overwhelmingly generated by language models. I suggest the authors render the texts more carefully. Some sentences I think are overly decorated..." - this is still an issue in parts of the manuscript, please revise carefully.

**Audience:**

Yes

**Audience Explanation:**

Reward design for RL is an important problem and ways to perform it more effectively are thus relevant to the respective subcommunity.

**Claims And Evidence:**

Yes

**Claims Explanation:**

The authors propose an approach for more effectively designing reward functions with the help of LLMs and demonstrate in experiments that this is indeed advantageous in comparison to baselines that leverage LLMs in different ways. Ablation studies are presented to highlight the key components of the proposed approach.

---

> ### Author Response · Authors · 2025-10-20
>
> We are grateful for the thorough review and constructive comments provided by the Associate Editor.
>
>
> Q1: What are the ''reward components''?
>
> A1: In Section 3 (Preliminary), we have added a definition of reward components, describing them as composable sub-terms that constitute the overall reward function, each corresponding to a specific behavioral metric or task-related objective. Moreover, illustrative examples are provided to facilitate understanding of this concept.
>
>
> Q2: The feedback available to the process.
>
> A2: We have clarified the specific contents of the feedback available to the process in Section 4.1 and added a detailed example in Appendix A.
>
>
> Q3: Does the definition of RDP not demand sample efficiency?
>
> A3: The original definition of RDP does not consider the constraint on sample efficiency. Yet, our work extends the standard RDP formulation by explicitly aiming to improve sample efficiency during the automatic reward design process.  We have clarified this in Section 3  (paragraph 3).
>
>
> Q4: RDP vs rdp and $U_{R_i}$
>
> A4: We have unified the notation for the policy space $\Pi$ in the definitions of RL and RDP in the camera-ready version. For ``$U_{R_i}$'', it is first introduced and defined in Section 4.2 (paragraph 3, line 10), where its meaning is explicitly explained. We have verified that all symbols are now clearly introduced upon their first appearance.
>
>
> Q5: ``Self-consistency''
>
> A5: ``Self-consistency'' is explained upon its first occurrence (Section 1, paragraph 3, line 4) together with an appropriate reference. Specifically, it refers to the phenomenon that LLMs exhibit higher output consistency when dealing with well-defined tasks.
>
>
> Q6: ``How to filter out potentially problematic reward function samples...''
>
> A6: We have revised the expression “problematic reward function samples” to ``redundant reward function samples'' for better precision and clarity.
>
>
> Q7: What is "optimal" supposed to mean?
>
> A7: We have revised the figure captions to clarify that the curves of all methods are truncated at their respective convergence points, where the performance becomes stable and reaches its optimal value. Thus, the term ``optimal performance'' refers to the best performance achieved before convergence.
>
>
> Q8: Writing expression.
>
> A8: In the camera-ready paper, we have thoroughly checked the manuscript to ensure that the writing is clear and easy to follow. Overly decorative expressions have been removed to improve readability.

---

> > ### Comment · Action_Editor_QNDK · 2025-11-09
> >
> > Thanks for your updates. I would want to ask for one last clarification. Regarding the plots you now say: "The URDP curve, as well as those of other methods, is truncated at the point where the performance reaches convergence and remains stable." Can you please clarify how these conditions are determined. It seems a bit surprising that that for all problems for Eureka and Text2Reward this convergence is reached at always ~80 evaluations while for URDP this happens at varying numbers of evaluations. I would also still argue that many of the curves very much don't look converged so it would be valuable to not truncate but rather show the continuation of the curves.
> > However, please clarify when exactly you truncate or continue all curves to the same number of evaluations.

---

> > > ### Author Response · Authors · 2025-11-21
> > >
> > > Thank you for your helpful request for clarification. We have redrawn the figures to ensure that all methods have reached convergence. In the updated figures, both URDP and the compared methods are considered to have converged when their HNS values no longer change after at least two consecutive outer iterations (i.e., the plot points in the curves). At this point, the inner loop (UABO) no longer yields better hyperparameters (as UABO produced identical results in three consecutive simulations), and the outer LLM-based optimization no longer improves the reward components. Therefore, further iterations are unnecessary. This convergence criterion is now explicitly stated in both the figure captions and the experimental setup. These updates clarify that all curves are extended until they reach stable optimal performance, rather than being truncated at a fixed number of evaluations.

---

> > > > ### Comment · Action_Editor_QNDK · 2025-11-21
> > > >
> > > > Thanks